# Uncover Underlying Correspondence for Robust Multi-view Clustering

**Haochen Zhou**[1]  **Guofeng Ding**[1]  **Mouxing Yang**[1]  **Peng Hu**[1]  **Yijie Lin**[1*]  **Xi Peng**[2,3*]

College of Computer Science, Sichuan University, China.[1]
School of Artificial Intelligence, Sichuan University, China.[2]
National Key Laboratory of Fundamental Algorithms and Models for Engineering
Numerical Simulation, Sichuan University, China.[3]
{haochenzhou.xl, guofengding.gm, yangmouxing, penghu.ml, linyijie.gm, pengx.gm}@gmail.com

## Abstract

Multi-view clustering (MVC) aims to group unlabeled data into semantically meaningful clusters by leveraging cross-view consistency. However, real-world datasets collected from the web often suffer from noisy correspondence (NC), which breaks the consistency prior and results in unreliable alignments. In this paper, we identify two critical forms of NC that particularly harm clustering: i) category-level mismatch, where semantically consistent samples from the same class are mistakenly treated as negatives; and ii) sample-level mismatch, where collected cross-view pairs are misaligned and some samples may even lack any valid counterpart. To address these challenges, we propose **CorreGen**, a generative framework that formulates noisy correspondence learning in MVC as maximum likelihood estimation over underlying cross-view correspondences. The objective is elegantly solved via an Expectation–Maximization algorithm: in the E-step, soft correspondence distributions are inferred across views, capturing class-level relations while adaptively down-weighting noisy or unalignable samples through GMM-guided marginals; in the M-step, the embedding network is updated to maximize the expected log-likelihood. Extensive experiments on both synthetic and real-world noisy datasets demonstrate that our method significantly improves clustering robustness. The code is available at https://github.com/XLearning-SCU/2026-ICLR-CorreGen.

## 1 Introduction

Describing the same object from multiple perspectives (Yan et al., 2021) or modalities (Sharma et al., 2018), multi-view data have become increasingly prevalent in real-world applications. To exploit such data, contrastive multi-view clustering (MVC) has emerged as a powerful unsupervised paradigm (Qin et al., 2025b; Wang et al., 2025a). Relying on the consistency prior that views from the same instance should be semantically aligned, contrastive MVC pulls positive pairs (*i.e.*, views of the same instance) closer while pushing negative pairs (*i.e.*, views from different instances) apart in the embedding space. Through this process, it could learn a shared embedding space across views and group unlabeled samples into semantically meaningful clusters.

However, this prior is often difficult to satisfy. In practice, multi-view datasets are commonly constructed by crawling paired data from the web, such as images with their associated alt text (Wang et al., 2015). This automatic process inevitably introduces the noisy correspondence (NC) problem (Huang et al., 2021), where cross-view pairs are incorrectly matched. Such noise undermines the cross-view consistency prior and severely distorts the semantic structure of the learned embedding space.

In this paper, we identify two major types of NC that are particularly harmful to clustering: i) *Category-level mismatch*, where views from different modalities but belonging to the same class are mistakenly treated as negatives by contrastive MVC methods, despite their underlying semantic

---

*Corresponding author.

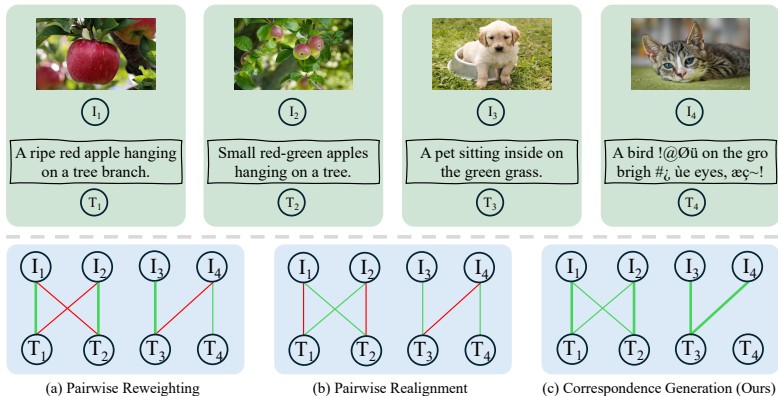

Figure 1: (*Top*) Examples of multi-view data, including noisy pairs $I_4$–$T_4$. (*Bottom*) Illustration of three paradigms for handling noisy correspondence, where green edges denote discovered correspondences and red edges indicate potential but undiscovered ones. (a) Pairwise reweighting, which applies robust contrastive losses to down-weight potentially noisy pairs during training but retains the original correspondences unchanged; (b) Pairwise realignment, which reassigns each sample to a more plausible cross-view counterpart; (c) Correspondence generation (Ours), which directly uncovers latent correspondences and filters out noise.

consistency; ii) *Sample-level mismatch*, which manifests in two scenarios: alignable mispairs, where a sample is wrongly paired with a mismatched view despite having a correct counterpart elsewhere; and unalignable samples, where no valid counterpart exists due to corruption, noise, or poor data quality. Such issues are especially prevalent in web-collected data, where the pairwise noise ratio can exceed 20% (Sharma et al., 2018; Wang et al., 2015). Critically, manually verifying or cleaning these correspondences is prohibitively expensive, underscoring the need for robust multi-view clustering methods. To address NC, recent works (Qu et al., 2025) mainly adopt either pairwise reweighting or realignment strategies, as illustrated in Fig. 1. However, both approaches overlook category-level semantics and unalignable samples, leading to suboptimal results in clustering.

Recognizing these limitations, we shift from the existing discriminative contrastive objective to a generative one. Specifically, we formulate noisy correspondence learning in MVC as a maximum likelihood estimation objective of the underlying joint distribution, in which the counterparts across views are modeled as unobserved latent variables. Unlike previous methods (Wang et al., 2025b; Qin et al., 2025a) that focus on verifying whether given positive or negative pairs are correctly aligned, our formulation uncovers the underlying correspondences without heavily relying on predefined (potentially noisy) pairs. By maximizing the overall log-likelihood, we capture the semantic structure in a principled and probabilistic manner.

To effectively optimize the proposed objective, we develop an Expectation-Maximization (EM) based algorithm **CorreGen**. In the E-step, the goal is to infer a latent correspondence distribution across views. We first estimate the marginal likelihood of each sample by fitting a Gaussian Mixture Model in the embedding space. Intuitively, this estimation assigns higher probabilities to samples that lie in large and coherent clusters, while noisy or unalignable samples receive lower probabilities. These marginals serve as constraints to solve an optimal transport formulation, yielding a soft many-to-many assignment that captures category-level relationships across views. In the M-step, the estimated correspondences are used to maximize the expected log-likelihood, updating the embedding network such that semantically consistent pairs are assigned higher likelihoods. Iterating between the two steps gradually uncovers reliable correspondences and refines robust cluster representations. In summary, the contribution of our work can be summarized as follows:

- We identify and formalize two types of noisy correspondence in MVC: category-level mismatch and sample-level mismatch, where both are prevalent in real-world multi-view datasets and harmful to clustering.

- We propose CorreGen, a novel generative framework that models latent cross-view correspondences through maximum likelihood estimation, solved elegantly via an EM algo-

rithm. Furthermore, we prove that the standard InfoNCE is a special case of our formulation under specific assumptions.

- We introduce a principled E-step solution that jointly models category-level correspondences and suppresses sample-level noise by leveraging GMM-guided marginals. Extensive experiments on both synthetic and real-world noisy datasets validate the effectiveness of our approach. Notably, our method achieves 10% accuracy improvements on the challenging UMPC-Food101 dataset (Wang et al., 2015).

## 2 RELATED WORK

**Robust Multi-view Clustering** aims to handle imperfections that commonly occur in real-world datasets. These imperfections can be broadly categorized into two types: i) *Incomplete Multi-view Problem* (IMP) arises when some views are missing, resulting in incomplete cross-view information. To mitigate this issue, recent methods adopt various completion-based strategies such as anchor learning (Liu et al., 2024), subspace learning (Zhang et al., 2024), or diffusion models (Zhang et al., 2025). These approaches aim to impute the missing views and recover complete multi-view representations; ii) *Partially view-aligned Problem* (PVP) occurs when the correspondences across views are misaligned. For example, in multi-camera surveillance, images of the same person from different cameras may be temporally asynchronous (Huang et al., 2020). To address this, He et al. (2024) introduce a variational contrastive learning framework to realign unpaired data, while Yan et al. (2025) design a multi-stage strategy that iteratively updates cross-view correspondences for unpaired data.

Although both PVP and NC address erroneous cross-view correspondences, the NC problem studied in this paper differs in two significant aspects. First, misalignments in NC are unobserved, with no manually verified labels or alignment indicators available (Lee et al., 2018). Second, NC encompasses not only instance-level mismatches, but also category-level misalignments and even unalignable samples that lack valid counterparts across views.

**Noisy Correspondence Learning** was first introduced in cross-modal retrieval (Huang et al., 2021), where mismatched image-text pairs are mistakenly treated as true positives. Recently, this problem has garnered increasing attention across a range of domains, including video reasoning (Lin et al., 2024), graph matching (Lin et al., 2023; 2026), person re-identification (Yang et al., 2022a) and multi-view clustering (Sun et al., 2024; 2025). Existing solutions can be broadly categorized into two groups: i) *Reweighting-based methods* (Yang et al., 2024) aim to reduce the impact of mismatched pairs by assigning them lower weights during training. For example, Huang et al. (2021) adjust the margins in triplet contrastive loss to account for false positives; ii) *Realignment-based methods* (Lin et al., 2024) attempt to reassign each sample to a more plausible counterpart across views, thereby mitigating alignment errors.

Although existing methods achieve promising results, they mainly refine given positive pairs while overlooking potential category-level correspondences, leading to suboptimal clustering performance. Different from these discriminative approaches, we propose a generative objective for noisy correspondence learning in MVC, which assigns higher likelihoods to semantically consistent samples and uncovers latent correspondences. Notably, our optimization does not rely heavily on off-the-shelf pairs, thereby mitigating the noisy correspondence problem from a new perspective.

## 3 METHOD

In this section, we first introduce the problem setting and formalize correspondence learning in multi-view clustering (MVC) as a generative maximum likelihood estimation problem in Sec. 3.1. To optimize this objective, we propose **CorreGen**, an EM-based framework in Sec. 3.2, and detail its two steps in Sec. 3.2.1 and Sec. 3.2.2.

### 3.1 PROBLEM DEFINITION

Given a multi-view dataset $\{(\boldsymbol{x}_i^{(1)}, \ldots, \boldsymbol{x}_i^{(V)})\}_{i=1}^N$ with $N$ instances observed from $V$ views, the goal of MVC is to learn an encoder $f_\theta$ that maps each view $\boldsymbol{x}_i^{(v)}$ into a shared embedding space,

*i.e.*, $\boldsymbol{z}_i^{(v)} = f_\theta(\boldsymbol{x}_i^{(v)})$. Ideally, the distribution of these embeddings should form $C$ well-separated semantic clusters, such that traditional clustering algorithms (*e.g.*, K-means (McQueen, 1967)) can easily distinguish them.

To achieve this goal, recent contrastive MVC methods (Yang et al., 2023) pull positive pairs (*i.e.*, views of the same instance) closer while pushing negative pairs (*i.e.*, views from different instances) apart in the embedding space. Formally, for any pair of views $(v_1, v_2)$ with $v_1 \neq v_2$, the positive and negative sets are defined as

$$\mathcal{P}_{v_1,v_2}^+ = \bigcup_{i=1}^{N} \{ (\boldsymbol{x}_i^{(v_1)}, \boldsymbol{x}_i^{(v_2)}, t_{ii}^{12} = 1)\}, \quad \mathcal{P}_{v_1,v_2}^- = \bigcup_{i=1}^{N} \bigcup_{j=1, j\neq i}^{N} \{ (\boldsymbol{x}_i^{(v_1)}, \boldsymbol{x}_j^{(v_2)}, t_{ij}^{12} = 0)\}, \quad (1)$$

where $t_{ij}^{12} \in \{0, 1\}$ is an indicator variable that equals 1 if $\boldsymbol{x}_i^{(v_1)}$ and $\boldsymbol{x}_j^{(v_2)}$ belong to the same instance, and 0 otherwise. Nevertheless, contrastive MVC essentially formulates an instance-level discriminative task (Wu et al., 2018), which overlooks the intrinsic cluster structure of data. As a result, real-world multi-view datasets are particularly vulnerable to the *noisy correspondence* problem, where the assumed cross-view alignment fails to hold. For clarity, we formalize its two manifestations, namely *category-level mismatch* and *sample-level mismatch*, as defined below.

**Definition 1** (Category-level mismatch). *Consider a cross-view pair $(\boldsymbol{x}_i^{(v_1)}, \boldsymbol{x}_j^{(v_2)}, t_{ij}^{12})$, where $t_{ij}^{12} \in \{0, 1\}$ denotes whether the pair is treated as positive or negative. Let $c_i^{(v_1)}$ and $c_j^{(v_2)}$ be the oracle class labels of $\boldsymbol{x}_i^{(v_1)}$ and $\boldsymbol{x}_j^{(v_2)}$, respectively. A category-level mismatch occurs if $c_i^{(v_1)} = c_j^{(v_2)}$ but $t_{ij}^{12} = 0$, i.e., samples from the same semantic class are incorrectly assigned as a negative pair.*

In other words, category-level mismatch occurs when semantically related instances are mistakenly treated as negatives. Ideally, all cross-view pairs of samples from the same class should be regarded as positives with $t_{ij}^{12} = 1$, rather than only those from the same instance.

**Definition 2** (Sample-level mismatch). *Consider a cross-view pair $(\boldsymbol{x}_i^{(v_1)}, \boldsymbol{x}_i^{(v_2)}, t_{ii}^{12})$, where $c_i^{(v_1)}$ and $c_i^{(v_2)}$ denote the oracle class labels of $\boldsymbol{x}_i^{(v_1)}$ and $\boldsymbol{x}_i^{(v_2)}$, respectively. A sample-level mismatch occurs if either i) $c_i^{(v_1)} \neq c_i^{(v_2)}$, or ii) at least one of $c_i^{(v_1)}$ or $c_i^{(v_2)}$ does not correspond to any valid class. In both cases, the pair cannot be regarded as a valid positive correspondence.*

Specifically, sample-level mismatch admits two scenarios: i) *alignable mispaired*: although the constructed pair is incorrect, the sample $\boldsymbol{x}_i^{(v_1)}$ still has a valid counterpart $\boldsymbol{x}_k^{(v_2)}$ in the other view. This case often co-occurs with category-level mismatch; ii) *unalignable mispaired*: there is no valid counterpart, *e.g.*, the sample $\boldsymbol{x}_i^{(v_1)}$ might be corrupted or purely noisy data.

These two types of complex noisy correspondence motivate a more fundamental question: can we reduce the reliance on pre-defined pairs and instead directly model the intrinsic relationships that couple different views? Building on this intuition, we adopt a generative formulation that maximizes the marginal log-likelihood of the observed multi-view data (Bengio et al., 2013):

$$\theta^* = \arg\max_\theta \sum_{v=1}^{V} \sum_{i=1}^{N} \log p(\boldsymbol{x}_i^{(v)}; \theta). \quad (2)$$

In multi-view clustering, each sample in one view may be associated with multiple counterparts in another view. Since these associations are unknown a priori, we treat them as latent variables. By aggregating over all unordered view pairs $(v_i, v_j)$, the objective can be reformulated as:

$$\theta^* = \arg\max_\theta \sum_{v_1}^{V} \sum_{i}^{N} \sum_{v_2}^{V} \log \sum_{j}^{N} p(\boldsymbol{x}_i^{(v_1)}, \boldsymbol{x}_j^{(v_2)}; \theta). \quad (3)$$

Maximizing this marginal likelihood implicitly encourages the model to learn a meaningful joint distribution $p(\boldsymbol{x}_i^{(v_1)}, \boldsymbol{x}_j^{(v_2)}; \theta)$. In particular, to maximize the inner summation over $j$, the parameters $\theta$ must assign higher joint probability to semantically consistent pairs, thereby revealing the underlying cross-view correspondences in a probabilistic sense.

Compared with discriminative objectives, this generative formulation offers two key advantages: i) it alleviates the heavy reliance on pre-defined positive and negative pairs, making it naturally robust to sample-level unmatchable cases; ii) it captures many-to-many probabilistic correspondences across views, which better reflects the complex coupling of real-world multi-view data and mitigates category-level mismatch. However, the nested summation in Eq. (3) makes direct optimization intractable. To address this, we cast the objective into the Expectation–Maximization (EM) framework and present the theoretical derivation in the next section.

## 3.2 Correspondence Generation via Expectation–Maximization

To simplify the derivation of the joint log-likelihood defined in Eq. (3), we first consider a subset of the objective involving only two views:

$$\theta^* = \arg\max_{\theta} \sum_{i=1}^{N} \log \sum_{j=1}^{N} p(\boldsymbol{x}_i^{(v_1)}, \boldsymbol{x}_j^{(v_2)}; \theta). \tag{4}$$

Directly optimizing Eq. (4) is intractable due to the nested log-sum over latent variables. To address this, we introduce an auxiliary distribution $Q(\boldsymbol{x}_j^{(v_2)})$ over $\boldsymbol{x}_j^{(v_2)}$ such that $\sum_{j=1}^{N} Q(\boldsymbol{x}_j^{(v_2)}) = 1$. This allows us to derive a lower bound:

$$\sum_{i=1}^{N} \log \sum_{j=1}^{N} p(\boldsymbol{x}_i^{(v_1)}, \boldsymbol{x}_j^{(v_2)}; \theta) = \sum_{i=1}^{N} \log \sum_{j=1}^{N} Q(\boldsymbol{x}_j^{(v_2)}) \frac{p(\boldsymbol{x}_i^{(v_1)}, \boldsymbol{x}_j^{(v_2)}; \theta)}{Q(\boldsymbol{x}_j^{(v_2)})}, \tag{5}$$

$$\geq \sum_{i=1}^{N} \sum_{j=1}^{N} Q(\boldsymbol{x}_j^{(v_2)}) \log \frac{p(\boldsymbol{x}_i^{(v_1)}, \boldsymbol{x}_j^{(v_2)}; \theta)}{Q(\boldsymbol{x}_j^{(v_2)})}, \tag{6}$$

where the inequality follows from Jensen's inequality. The bound becomes tight when $Q(\boldsymbol{x}_j^{(v_2)}) = p(\boldsymbol{x}_j^{(v_2)}; \boldsymbol{x}_i^{(v_1)}, \theta)$, *i.e.*, when the auxiliary distribution matches the posterior under the current parameters $\theta^{(t)}$. Substituting this choice of $Q$ into the bound gives:

$$\theta^* = \arg\max_{\theta} \sum_{i=1}^{N} \sum_{j=1}^{N} Q(\boldsymbol{x}_j^{(v_2)}) \log p(\boldsymbol{x}_i^{(v_1)}, \boldsymbol{x}_j^{(v_2)}; \theta) - \sum_{i=1}^{N} \sum_{j=1}^{N} Q(\boldsymbol{x}_j^{(v_2)}) \log Q(\boldsymbol{x}_j^{(v_2)}) \tag{7}$$

$$= \arg\max_{\theta} \sum_{i=1}^{N} \sum_{j=1}^{N} p(\boldsymbol{x}_j^{(v_2)}; \boldsymbol{x}_i^{(v_1)}, \theta^{(t)}) \log p(\boldsymbol{x}_i^{(v_1)}, \boldsymbol{x}_j^{(v_2)}; \theta), \tag{8}$$

where the entropy term $-\sum_i^{N} \sum_j^{N} Q(\boldsymbol{x}_j^{(v_2)}) \log Q(\boldsymbol{x}_j^{(v_2)})$ is omitted since it is independent of $\theta$. In the **E-step**, we estimate the posterior distribution $p(\boldsymbol{x}_j^{(v_2)}; \boldsymbol{x}_i^{(v_1)}, \theta^{(t)})$, which provides a soft assignment of correspondences between samples across views. In the **M-step**, we maximize the weighted log-likelihood in Eq. (8), updating the parameters $\theta$ guided by the correspondences inferred in the E-step. By aggregating over all views, the above derivation naturally generalizes to multiple views. Fig. 2 shows an overview of the above EM process and the details of the two steps will be discussed in the next section.

### 3.2.1 E-step: Estimating Underlying Correspondences

In the E-step, we estimate the posterior distribution of latent correspondences $p(\boldsymbol{x}_j^{(v_2)}; \boldsymbol{x}_i^{(v_1)}, \theta^{(t)})$ under the current parameters $\theta^{(t)}$:

$$p(\boldsymbol{x}_j^{(v_2)}; \boldsymbol{x}_i^{(v_1)}, \theta^{(t)}) = \frac{p(\boldsymbol{x}_i^{(v_1)}, \boldsymbol{x}_j^{(v_2)}; \theta^{(t)})}{p(\boldsymbol{x}_i^{(v_1)}; \theta^{(t)})}, \tag{9}$$

which naturally decomposes the estimation into two parts, namely, the marginal distribution of individual views and the joint distribution across views.

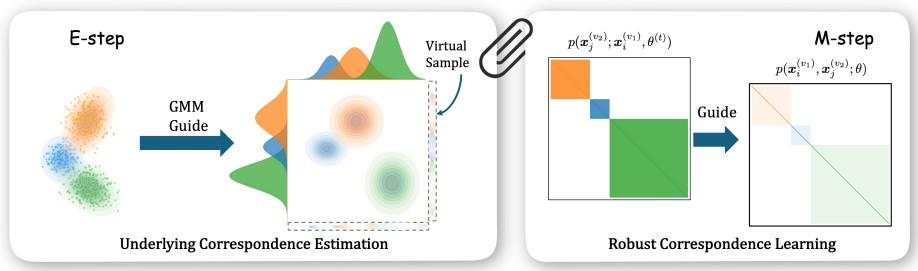

Figure 2: Overview of the CorreGen framework which operates via an EM procedure: the E-step infers the underlying correspondence distribution using GMM-guided marginals and a virtual sample mechanism to handle noise; the M-step subsequently utilizes these estimated soft correspondences to guide the robust representation learning.

First, we estimate the joint distribution between views $v_1$ and $v_2$, represented as a matrix $\boldsymbol{P} \in \mathbb{R}_+^{N \times N}$ where each entry $\boldsymbol{P}_{ij} = p(\boldsymbol{x}_i^{(v_1)}, \boldsymbol{x}_j^{(v_2)}; \theta^{(t)})$. A good estimate of $\boldsymbol{P}$ should not only satisfy the marginal constraints but also capture the semantic dependency between the two views. Inspired by recent works that utilize Optimal Transport (OT) to model complex cross-view relationships for clustering (Deng et al., 2025; Fu et al., 2025), we formulate the estimation of the optimal joint distribution as an OT problem. We introduce a correlation function $s(\boldsymbol{z}_i^{(v_1)}, \boldsymbol{z}_j^{(v_2)})$ (e.g. cosine similarity) to measure the semantic correlations of a sample pair under the current parameters $\theta^{(t)}$, with $\boldsymbol{z}_i^{(v)} = f_{\theta^{(t)}}(\boldsymbol{x}_i^{(v)})$. Then the expected correlation is defined as

$$\mathbb{E}_{\boldsymbol{P}}[s] = \sum_{i=1}^{N} \sum_{j=1}^{N} \boldsymbol{P}_{ij} \, s(\boldsymbol{z}_i^{(v_1)}, \boldsymbol{z}_j^{(v_2)}). \tag{10}$$

We then seek the optimal joint distribution by maximizing this expectation:

$$\boldsymbol{P}^* = \underset{\boldsymbol{P} \in \Pi(\boldsymbol{p}^{(v_1)}, \boldsymbol{p}^{(v_2)})}{\arg\max} \mathbb{E}_{\boldsymbol{P}}[s]$$

$$\text{s.t} \quad \Pi(\boldsymbol{p}^{(v_1)}, \boldsymbol{p}^{(v_2)}) = \left\{ \boldsymbol{P} \in \mathbb{R}_+^{N \times N} \middle| \boldsymbol{P} \boldsymbol{1}_N = \boldsymbol{p}^{(v_1)}, \boldsymbol{P}^\top \boldsymbol{1}_N = \boldsymbol{p}^{(v_2)} \right\}, \tag{11}$$

where $\boldsymbol{p}^{(v_i)}$ is the marginal distribution vector for view $v_i$, i.e., $\boldsymbol{p}_i^{(v_i)} = p(\boldsymbol{x}_i^{(v_i)}; \theta^{(t)})$. This formulation ensures that the estimated joint distribution preserves the marginal constraints while assigning higher probability mass to semantically correlated pairs. However, due to the sample-level unalignable problem, there may exist outliers whose joint probabilities with all other samples should ideally be close to zero.

**Virtual Sample for Partial Alignment.** To handle the outliers and obtain a more realistic joint distribution, we first introduce a virtual sample for each view to represent the outliers. Let $\rho$ denote the potential noise ratio, which corresponds to the marginal probability mass of the virtual sample. We then augment the joint distribution to $\tilde{\boldsymbol{P}} \in \mathbb{R}_+^{(N+1) \times (N+1)}$, ensuring that the total probability mass assigned to outliers equals $\rho$. Formally, $\tilde{\boldsymbol{P}}$ satisfies

$$\tilde{\boldsymbol{P}} \boldsymbol{1}_{N+1} = [\boldsymbol{p}^{(v_1)}; \rho], \quad \tilde{\boldsymbol{P}}^\top \boldsymbol{1}_{N+1} = [\boldsymbol{p}^{(v_2)}; \rho], \tag{12}$$

which enables the model to absorb unalignable or noisy samples into the virtual probability mass.

Recall from Eq. (9) that estimating the posterior probabilities requires both the joint distribution $p(\boldsymbol{x}_i^{(v_1)}, \boldsymbol{x}_j^{(v_2)}; \theta^{(t)})$ and the marginal distribution $p(\boldsymbol{x}_i^{(v_1)}; \theta^{(t)})$. In the expectation formulation Eq. (11), these marginals act as constraints on the feasible set of couplings $\Pi(\boldsymbol{p}^{(v_1)}, \boldsymbol{p}^{(v_2)})$, which essentially determines how many valid counterparts each sample can align with. Under category-level mismatch, the number of valid counterparts is not uniform but depends on the size and structure of the sample's semantic class. Therefore, the marginal distribution should naturally reflect this variability: *samples from larger clusters or closer to cluster centers are assigned higher alignment mass, while outliers receive lower probabilities.*

**GMM-guided Marginal Estimation.** We assume that each sample is generated from a latent semantic cluster, which can be approximated by an anisotropic Gaussian distribution $\boldsymbol{x}_i^{(v)} \sim \mathcal{N}(\boldsymbol{\mu}_c, \boldsymbol{\Sigma}_c)$. Accordingly, we fit the embedding space of each view with a Gaussian Mixture Model (GMM) and compute the posterior responsibility of each cluster for every sample. The marginal probability is then estimated as

$$p(\boldsymbol{x}_i^{(v)}; \theta^{(t)}) = \frac{m^{d_i} - 1}{m - 1} \cdot \frac{N_c}{N},$$

(13)

$$d_i = \exp\left(-\epsilon\sqrt{(\boldsymbol{z}_i^{(v)} - \boldsymbol{\mu}_c)^\top \boldsymbol{\Sigma}_c^{-1}(\boldsymbol{z}_i^{(v)} - \boldsymbol{\mu}_c)}\right),$$

(14)

where $N_c$ is the number of samples assigned to cluster $c$ by GMM, $\epsilon$ and $m$ are shaping parameters. Concretely, we first compute the Mahalanobis distance in Eq. (14) between each sample and its cluster center, and map the result through an exponential kernel to obtain an assignment confidence $d_i$. This confidence is further passed through a curve-shaping function $\frac{m^{d_i}-1}{m-1}$, which amplifies the contrast between high- and low-confidence samples: samples closer to the cluster center receive disproportionately higher weights, while distant ones are smoothly down-weighted rather than suppressed abruptly. Finally, the re-scaled confidence is combined with the cluster proportion $N_c/N$ to yield the final probability to fill the marginal distribution in Eq. (11). In practice, we set $\epsilon = 0.1$ and $m = 10$, and apply a momentum update to stabilize training.

**Proposition 1.** *Eq. (11) with virtual sample can be solved by an efficient scaling algorithm if adding an entropy regularization $\lambda\mathcal{H}(\tilde{\boldsymbol{P}})$, where $\lambda$ is a regularization factor. Specifically, we derive the optimal augmented joint distribution $\tilde{\boldsymbol{P}}^*$ through the following iterations:*

$$\tilde{\boldsymbol{P}}^* = Diag(\boldsymbol{u}) \exp(\tilde{\boldsymbol{S}}/\lambda) Diag(\boldsymbol{v}),$$

$$\text{with iteration update} \quad \boldsymbol{u} \leftarrow \tilde{\boldsymbol{p}}^{(v_1)}/(\exp(\tilde{\boldsymbol{S}}/\lambda)\boldsymbol{v}), \quad \boldsymbol{v} \leftarrow \tilde{\boldsymbol{p}}^{(v_2)}/(\exp(\tilde{\boldsymbol{S}}^\top/\lambda)\boldsymbol{u}).$$

(15)

*where $\boldsymbol{u} \in \mathbb{R}_+^{N+1}$, $\boldsymbol{v} \in \mathbb{R}_+^{N+1}$ are two scaling vectors, $\tilde{\boldsymbol{p}}^{(v_i)} = [\boldsymbol{p}^{(v_i)}; \rho]$. The extended correlation matrix $\tilde{\boldsymbol{S}} \in \mathbb{R}^{(N+1)\times(N+1)}$ is constructed as:*

$$\tilde{\boldsymbol{S}} = \begin{bmatrix} \boldsymbol{S} & \boldsymbol{0}_{N\times 1} \\ \boldsymbol{0}_{1\times N} & A \end{bmatrix},$$

(16)

*where $\boldsymbol{S}_{ij} = s(\boldsymbol{z}_i^{(v_1)}, \boldsymbol{z}_j^{(v_2)})$ and $A$ is a constant. The optimal joint distribution estimation $\boldsymbol{P}^*$ is obtained by discarding the last row and column of $\tilde{\boldsymbol{P}}^*$, i.e., $\boldsymbol{P}^* = \tilde{\boldsymbol{P}}^*_{1:N,1:N}$. The proof is provided in Appendix A.*

### 3.2.2 M-STEP: ROBUST CORRESPONDENCE LEARNING

In the M-step, we maximize the overall log-likelihood of the observed data based on the estimated posterior distribution. To make Eq. (8) tractable, we approximate the joint distribution $p(\boldsymbol{x}_i^{(v_1)}, \boldsymbol{x}_j^{(v_2)}; \theta)$ by normalizing the similarity scores of embeddings in the latent space

$$p(\boldsymbol{x}_i^{(v_1)}, \boldsymbol{x}_j^{(v_2)}; \theta) = \frac{\exp(s(\boldsymbol{z}_i^{(v_1)}, \boldsymbol{z}_j^{(v_2)})/\tau)}{\sum_{m=1}^{N}\sum_{n=1}^{N}\exp(s(\boldsymbol{z}_m^{(v_1)}, \boldsymbol{z}_n^{(v_2)})/\tau)},$$

(17)

where $\boldsymbol{z}_i^{(v)} = f_\theta(\boldsymbol{x}_i^{(v)})$ denotes the embedding of $\boldsymbol{x}_i^{(v)}$ and $\tau$ is a temperature parameter. According to Eq. (9), we compute the posterior using the optimal joint distribution $\boldsymbol{P}^*$ and marginals $\boldsymbol{p}^{(v_1)}$ obtained in the E-step, defined as $\boldsymbol{Q}_{ij} = \boldsymbol{P}_{ij}^*/\boldsymbol{p}_i^{(v_1)}$. Substituting this parameterization into Eq. (8), the M-step objective becomes

$$\theta^* = \arg\max_\theta \sum_{i=1}^{N}\sum_{j=1}^{N} \boldsymbol{Q}_{ij} \log \frac{\exp(s(\boldsymbol{z}_i^{(v_1)}, \boldsymbol{z}_j^{(v_2)})/\tau)}{\sum_{m=1}^{N}\sum_{n=1}^{N}\exp(s(\boldsymbol{z}_m^{(v_1)}, \boldsymbol{z}_n^{(v_2)})/\tau)},$$

(18)

where $s(\cdot, \cdot)$ denotes a correlation function. Unlike contrastive objectives that rely on manually defined positive/negative pairs, this formulation leverages the soft correspondences $\boldsymbol{P}^*$ inferred in the E-step, thereby mitigating the negative effects of noisy correspondence and enabling more robust representation learning. Importantly, we find that the widely used InfoNCE loss can be unified into our framework as a special case as stated below.

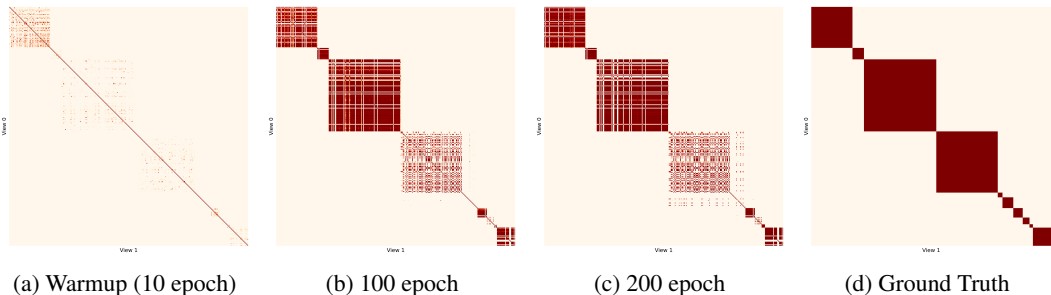

| (a) Warmup (10 epoch) | (b) 100 epoch | (c) 200 epoch | (d) Ground Truth |

Figure 3: Estimated posterior distributions over the course of training on the Caltech101 dataset.

**Proposition 2.** *If the marginal distribution $p(\boldsymbol{x}_i^{(v)}; \theta)$ is uniform and the posterior probability degenerates to $p(\boldsymbol{x}_i^{(v_2)}; \boldsymbol{x}_i^{(v_1)}, \theta) = 1$ (i.e., only paired cross-view samples are treated as valid positives), then Eq. (8) reduces to the standard InfoNCE contrastive objective:*

$$\theta^* = \arg\max_\theta \sum_{i=1}^N \log \frac{\exp(s(\boldsymbol{z}_i^{(v_1)}, \boldsymbol{z}_i^{(v_2)})/\tau)}{\sum_{n=1}^N \exp(s(\boldsymbol{z}_i^{(v_1)}, \boldsymbol{z}_n^{(v_2)})/\tau)}. \tag{19}$$

*The proof is in Appendix B.*

## 4 EXPERIMENTS

In this section, we conduct extensive experiments to evaluate the effectiveness of our method in addressing both category-level and sample-level noisy correspondence. Our study is guided by the following research questions: **Q1**: Does our method outperform existing robust MVC approaches under noisy correspondence (Section 4.2)? **Q2**: Can our method reliably uncover underlying category-level correspondences across views (Section 4.3)? **Q3**: How does performance vary under different levels of mismatch (Appendix D)? **Q4**: How sensitive is our method to hyperparameter choices (Appendix E)? **Q5**: Are the proposed components crucial for the improvements (Appendix F)?

### 4.1 EXPERIMENTAL SETUP

**Datasets.** We evaluate our method on four widely used datasets: Scene15 (Fei-Fei & Perona, 2005), Caltech101 (Li et al., 2015), LandUse21 (Yang & Newsam, 2010), and UMPC-Food101 (Wang et al., 2015). Notably, UMPC-Food101 contains images from 101 food categories paired with recipes crawled from the web, which inevitably introduces substantial irrelevant or noisy information. Representative examples of such noisy image–text pairs are provided in Appendix I.

**Baselines.** We compare CorreGen against seven state-of-the-art MVC methods, including DCP (Lin et al., 2022), SURE (Yang et al., 2022b), GCFAgg (Yan et al., 2023), CGCN (Wang et al., 2024), DIVIDE (Lu et al., 2024), CANDY (Guo et al., 2024), and ROLL (Sun et al., 2025). For fair comparison, we apply a view realignment strategy to the learned representations following prior studies (Guo et al., 2024; Sun et al., 2025), where realignment is consistently performed within batches of 512 to ensure fair evaluation.

**Implementation Details.** CorreGen introduces a generative objective for MVC that can be seamlessly integrated into existing contrastive frameworks. We implement it on top of DIVIDE (Lu et al., 2024) as the base model. More details are provided in Appendix C.

### 4.2 PERFORMANCE COMPARISON (Q1)

Since MVC is an unsupervised task, category-level correspondences depend on the underlying class sizes and distributions, making category-level mismatch an intrinsic challenge rather than one that can be explicitly specified. Therefore, in this section, we focus on evaluating model performance under different *sample-level mismatch* settings, which include two cases: i) *alignable mispairs*, caused by instance-level permutations across views; and ii) *unalignable mispairs*, caused by noisy

Table 1: The clustering performance with different mismatch ratios (MR). The best results and second best results are marked in **bold** and underline. All the results are the mean of five individual runs with different random seeds.

| MR Ratio | Method | Scene15 | | | LandUse21 | | | Caltech101 | | | UMPC-Food101 | | |
|---|---|---|---|---|---|---|---|---|---|---|---|---|---|
| | | ACC | NMI | ARI | ACC | NMI | ARI | ACC | NMI | ARI | ACC | NMI | ARI |
| 0% | DCP | 40.16 | 42.71 | 23.00 | 24.20 | 30.88 | 11.70 | 51.91 | 74.91 | 47.57 | 16.33 | 36.56 | 7.50 |
| | SURE | 43.41 | 44.33 | 25.71 | 23.14 | 29.20 | 10.62 | 38.94 | 65.64 | 27.28 | 29.86 | 46.37 | 19.22 |
| | GCFAgg | 33.58 | 32.91 | 16.76 | 23.48 | 26.75 | 10.80 | 34.27 | 55.57 | 19.98 | 16.12 | 30.03 | 6.55 |
| | CGCN | 41.34 | 40.09 | 24.64 | 23.57 | 26.88 | 10.40 | 36.40 | 66.72 | 24.72 | 29.58 | 39.57 | 14.69 |
| | DIVIDE | 44.57 | 45.98 | 28.43 | 32.50 | 39.44 | 18.16 | 62.20 | 83.30 | 50.50 | 36.20 | 57.92 | 27.72 |
| | CANDY | 42.55 | 41.67 | 25.41 | 30.94 | 36.33 | 16.20 | 67.64 | 84.06 | 60.02 | 33.10 | 53.06 | 22.39 |
| | ROLL | 47.61 | 48.71 | 30.86 | 29.43 | 33.78 | 15.24 | 17.83 | 42.75 | 13.43 | 23.65 | 47.22 | 16.43 |
| | Ours | 50.25 | 48.92 | 32.87 | 32.87 | 39.52 | 18.54 | 68.52 | 84.45 | 63.45 | 49.77 | 58.36 | 35.73 |
| 20% | DCP | 35.88 | 37.63 | 16.51 | 24.20 | 28.46 | 10.10 | 43.99 | 70.83 | 35.43 | 17.83 | 35.63 | 8.45 |
| | SURE | 37.26 | 35.56 | 19.94 | 24.67 | 27.45 | 10.91 | 35.91 | 60.06 | 24.56 | 20.30 | 32.89 | 8.99 |
| | GCFAgg | 33.11 | 27.64 | 15.29 | 23.86 | 23.30 | 9.11 | 28.90 | 47.47 | 13.81 | 11.28 | 19.48 | 2.94 |
| | CGCN | 35.96 | 35.73 | 20.10 | 24.52 | 26.38 | 10.36 | 33.01 | 64.17 | 24.41 | 28.01 | 38.36 | 13.63 |
| | DIVIDE | 41.91 | 40.16 | 24.84 | 30.89 | 35.93 | 16.21 | 55.65 | 70.72 | 50.92 | 31.41 | 51.21 | 22.70 |
| | CANDY | 41.05 | 40.41 | 24.44 | 30.54 | 35.45 | 15.99 | 65.79 | 82.29 | 60.03 | 30.41 | 50.36 | 20.36 |
| | ROLL | 44.86 | 46.96 | 28.71 | 29.33 | 33.11 | 15.16 | 20.39 | 46.44 | 15.03 | 21.26 | 43.05 | 13.73 |
| | Ours | 48.04 | 47.36 | 30.75 | 32.26 | 38.76 | 17.83 | 68.01 | 84.23 | 62.78 | 46.76 | 55.22 | 32.46 |
| 50% | DCP | 25.28 | 25.24 | 5.78 | 24.01 | 26.95 | 8.37 | 41.52 | 69.35 | 29.59 | 13.36 | 24.04 | 4.60 |
| | SURE | 28.16 | 26.52 | 13.16 | 22.67 | 24.91 | 9.94 | 26.89 | 52.51 | 18.73 | 11.06 | 21.51 | 3.20 |
| | GCFAgg | 21.07 | 11.26 | 5.14 | 24.48 | 22.56 | 8.92 | 22.16 | 36.65 | 8.89 | 6.70 | 11.02 | 0.80 |
| | CGCN | 35.99 | 33.07 | 19.47 | 20.62 | 23.35 | 7.83 | 37.74 | 65.66 | 28.20 | 20.71 | 31.44 | 8.51 |
| | DIVIDE | 39.67 | 36.47 | 22.69 | 29.75 | 33.17 | 15.23 | 38.81 | 59.18 | 33.03 | 25.21 | 44.47 | 16.00 |
| | CANDY | 41.25 | 39.02 | 23.93 | 29.09 | 32.56 | 14.77 | 60.30 | 78.60 | 55.16 | 28.80 | 48.69 | 19.03 |
| | ROLL | 42.41 | 44.49 | 26.43 | 28.65 | 32.81 | 15.01 | 18.57 | 43.50 | 13.68 | 20.97 | 38.54 | 11.89 |
| | Ours | 45.07 | 44.97 | 27.87 | 32.03 | 37.98 | 17.84 | 66.60 | 83.61 | 62.38 | 42.57 | 51.79 | 27.29 |
| 80% | DCP | 21.46 | 21.15 | 2.87 | 21.17 | 22.59 | 7.17 | 32.13 | 58.16 | 20.78 | 12.31 | 20.48 | 4.05 |
| | SURE | 24.57 | 23.68 | 9.90 | 17.57 | 19.61 | 5.94 | 23.61 | 49.01 | 15.97 | 8.81 | 18.32 | 2.19 |
| | GCFAgg | 11.53 | 3.08 | 0.90 | 17.38 | 15.17 | 4.44 | 16.61 | 32.57 | 5.78 | 3.58 | 6.90 | 0.14 |
| | CGCN | 28.81 | 25.42 | 12.89 | 20.29 | 20.70 | 7.32 | 35.32 | 63.83 | 25.77 | 18.13 | 29.48 | 6.92 |
| | DIVIDE | 35.90 | 32.95 | 19.63 | 28.56 | 31.74 | 14.32 | 27.42 | 53.68 | 21.56 | 24.78 | 42.98 | 15.63 |
| | CANDY | 38.27 | 36.08 | 20.74 | 28.44 | 31.39 | 14.01 | 54.17 | 77.30 | 53.79 | 27.59 | 48.10 | 17.62 |
| | ROLL | 37.62 | 38.27 | 21.19 | 25.67 | 28.42 | 11.96 | 20.83 | 45.58 | 13.97 | 19.39 | 39.68 | 13.52 |
| | Ours | 40.96 | 41.74 | 24.74 | 31.52 | 37.21 | 17.75 | 64.74 | 82.77 | 61.78 | 43.00 | 53.03 | 27.12 |

or corrupted samples. We control these two factors using the Mismatch Ratio (MR) and Corruption Ratio (CR), with detailed construction described in Appendix C.

Table 1 reports results under different MR. Our method consistently achieves the best performance, benefiting from its generative objective and robust correspondence discovery, which remain effective even with a few aligned pairs. Table 2 further evaluates scenarios with both alignable and unalignable mismatches. While all baselines degrade severely as MR and CR increase, our method maintains strong performance by jointly leveraging GMM-based marginals to down-weight noisy samples and virtual samples to absorb unalignable ones, mitigating the influence of low-quality pairs.

### 4.3 POSTERIOR DISTRIBUTION VISUALIZATION (Q2)

We next investigate whether CorreGen can uncover the latent correspondences across views. On Caltech101 with MR=0.2 and CR=0.0, we sample a mini-batch and estimate their posterior distributions at different training stages, comparing them with the true category-level ground truth.

As shown in Fig. 3, the category-level correlations are weak in the early training phase. By mid-training, the estimated posterior distributions already resemble the ground truth, and the gap further narrows in the later stages. These results demonstrate that CorreGen progressively uncovers the latent class-level correspondences, thereby effectively alleviating category-level mismatches.

## 5 CONCLUSION

In this paper, we propose a novel generative framework for multi-view clustering under the noisy correspondence challenge. Unlike existing discriminative approaches that rely heavily on off-the-shelf pairwise alignments, our method models cross-view dependencies by maximizing the joint likelihood of observed data, thereby uncovering latent correspondences in a principled manner. Extensive experiments across multiple datasets demonstrate that our approach not only achieves supe-

Table 2: The clustering performance on four multi-view datasets with different Mismatch Ratios (MR) and Corruption Ratios (CR).

| Setting | Method | Scene15 | | | LandUse21 | | | Caltech101 | | | UMPC-Food101 | | |
|---|---|---|---|---|---|---|---|---|---|---|---|---|---|
| | | ACC | NMI | ARI | ACC | NMI | ARI | ACC | NMI | ARI | ACC | NMI | ARI |
| MR 0.2 CR 0.2 | DCP | 36.50 | 40.52 | 21.55 | 24.62 | 29.19 | 11.37 | 43.03 | 69.34 | 37.81 | 12.97 | 28.99 | 4.71 |
| | SURE | 37.93 | 38.53 | 21.23 | 24.48 | 28.32 | 11.02 | 33.71 | 58.99 | 20.69 | 13.14 | 25.66 | 4.95 |
| | GCFAgg | 29.59 | 26.33 | 14.22 | 24.29 | 25.13 | 10.70 | 28.57 | 45.65 | 14.21 | 8.89 | 17.07 | 2.11 |
| | CGCN | 27.78 | 26.95 | 12.92 | 23.52 | 23.96 | 8.81 | 35.61 | 64.81 | 30.16 | 28.02 | 39.04 | 13.57 |
| | DIVIDE | 36.05 | 36.18 | 20.22 | 29.30 | 34.69 | 15.13 | 56.13 | 73.31 | 53.82 | 29.01 | 49.69 | 20.92 |
| | CANDY | 35.57 | 37.00 | 20.71 | 29.13 | 33.70 | 14.87 | 65.80 | 82.23 | 62.52 | 30.13 | 49.77 | 20.06 |
| | ROLL | 36.13 | 36.76 | 17.99 | 23.15 | 24.28 | 8.39 | 16.50 | 40.44 | 12.16 | 18.51 | 39.78 | 11.63 |
| | Ours | **41.23** | **41.43** | **25.05** | **31.13** | **37.36** | **17.00** | **67.12** | **84.45** | **64.13** | **45.97** | **54.66** | **31.36** |
| MR 0.2 CR 0.5 | DCP | 34.31 | 37.70 | 19.55 | 17.95 | 22.13 | 5.96 | 36.98 | 63.14 | 32.46 | 7.36 | 17.71 | 1.58 |
| | SURE | 34.05 | 35.32 | 18.37 | 20.05 | 23.20 | 7.40 | 32.18 | 58.49 | 20.47 | 6.86 | 15.83 | 4.19 |
| | GCFAgg | 27.85 | 24.05 | 12.73 | 23.24 | 24.19 | 9.92 | 27.57 | 45.00 | 14.43 | 7.77 | 15.68 | 1.67 |
| | CGCN | 28.36 | 31.46 | 16.32 | 22.24 | 25.04 | 9.61 | 35.83 | 76.99 | 41.69 | 24.07 | 35.01 | 10.17 |
| | DIVIDE | 33.54 | 35.40 | 19.90 | 27.94 | 31.81 | 13.75 | 57.87 | 76.59 | 58.56 | 24.92 | 46.78 | 17.61 |
| | CANDY | 31.24 | 34.08 | 19.00 | 24.72 | 28.03 | 11.27 | 62.57 | 81.52 | 55.76 | 25.00 | 47.27 | 17.36 |
| | ROLL | 27.03 | 25.83 | 9.42 | 16.40 | 15.49 | 3.20 | 12.97 | 36.57 | 9.80 | 16.12 | 36.52 | 9.66 |
| | Ours | **36.48** | 37.66 | **21.14** | **28.50** | **33.09** | **14.31** | 61.19 | **82.15** | 49.65 | **43.54** | **53.66** | **29.07** |
| MR 0.5 CR 0.2 | DCP | 33.62 | 35.05 | 14.48 | 24.48 | 27.57 | 10.35 | 38.03 | 64.81 | 30.53 | 9.30 | 19.71 | 2.44 |
| | SURE | 25.37 | 26.07 | 11.48 | 21.38 | 24.14 | 8.08 | 27.52 | 53.57 | 15.64 | 6.86 | 15.83 | 1.58 |
| | GCFAgg | 24.26 | 13.31 | 6.45 | 22.00 | 19.02 | 7.77 | 23.83 | 38.62 | 10.43 | 5.24 | 9.64 | 0.57 |
| | CGCN | 29.65 | 29.89 | 15.37 | 23.57 | 24.86 | 9.08 | 29.22 | 58.19 | 26.19 | 25.08 | 35.71 | 11.60 |
| | DIVIDE | 32.88 | 32.87 | 18.08 | 29.00 | 32.49 | 14.37 | 43.98 | 61.51 | 37.87 | 23.04 | 43.28 | 14.71 |
| | CANDY | 34.60 | 35.31 | 19.84 | 27.77 | 31.46 | 13.63 | 58.35 | 78.55 | 56.14 | 27.97 | 48.24 | 18.81 |
| | ROLL | 35.23 | 35.79 | 18.54 | 23.34 | 23.99 | 8.83 | 14.78 | 38.46 | 11.07 | 17.54 | 35.48 | 9.67 |
| | Ours | **39.54** | **39.55** | **23.12** | **31.20** | **36.25** | **16.92** | **66.87** | **84.15** | **67.31** | **38.84** | **50.09** | **24.98** |
| MR 0.5 CR 0.5 | DCP | 26.35 | 31.84 | 13.42 | 18.52 | 23.32 | 7.40 | 32.34 | 58.43 | 21.55 | 5.19 | 10.86 | 0.54 |
| | SURE | 26.91 | 28.73 | 12.06 | 19.57 | 21.18 | 6.60 | 25.90 | 54.83 | 18.07 | 7.00 | 17.28 | 1.77 |
| | GCFAgg | 22.27 | 14.13 | 6.68 | 20.57 | 17.30 | 6.72 | 21.56 | 37.88 | 9.61 | 4.61 | 8.88 | 0.42 |
| | CGCN | 27.27 | 30.11 | 14.68 | 19.67 | 22.51 | 7.38 | 33.15 | 59.86 | 24.95 | 20.74 | 32.53 | 8.41 |
| | DIVIDE | 30.27 | 31.25 | 16.31 | 26.13 | 29.12 | 12.30 | 48.07 | 68.23 | 44.69 | 20.67 | 42.07 | 12.52 |
| | CANDY | 29.44 | 32.67 | 17.09 | 24.08 | 27.21 | 11.01 | 51.28 | 75.16 | 41.70 | 24.70 | 46.58 | 17.19 |
| | ROLL | 26.29 | 24.98 | 9.41 | 14.62 | 13.00 | 2.19 | 13.82 | 36.54 | 10.30 | 14.76 | 32.84 | 7.71 |
| | Ours | **36.19** | **36.84** | **20.83** | **28.72** | **32.54** | **14.50** | **57.06** | **80.34** | **45.37** | **37.26** | **49.30** | **23.25** |

rior clustering performance but also exhibits strong robustness to sample-level and category-level mismatches. In the future, we plan to extend this framework to unpaired multi-modal learning and apply it to cross-modal retrieval tasks with large-scale noisy data.

## AUTHOR CONTRIBUTIONS

All authors contributed significantly to this work. Xi Peng and Yijie Lin conceived the study, designed the CorreGen algorithm, and refined the manuscript. Haochen Zhou co-designed and implemented the CorreGen algorithm, conducted the experiments, and drafted the manuscript. Guofeng Ding supplemented the baseline evaluations and contributed to the content of the manuscript. Peng Hu and Mouxing Yang provided suggestions for the algorithm's design and contributed to the formulation of the manuscript. All authors reviewed and approved the final version.

## ACKNOWLEDGMENTS

This work was supported in part by the Fundamental Research Funds for the Central Universities under Grant CJ202303, CJ202403; in part by NSFC under Grant 624B2099, U25A201523, 62472295; in part by Sichuan Science and Technology Planning Project under Grant 24NSFTD0130; in part by Fundamental and Interdisciplinary Disciplines Breakthrough Plan of the Ministry of Education of China under Grant JYB2025XDXM610; in part by System of Systems and Artificial Intelligence Laboratory pioneer fund grant under Grant HLJGGG20240327517-15.

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

APPENDIX

## A    EFFICIENT SOLVER FOR JOINT DISTRIBUTION ESTIMATION (PROOF OF PROPOSITION 1)

In this section, we derive the efficient solver for Eq. (11) under the virtual-sample formulation, where the marginals are augmented with a virtual probability mass $\rho$ following Eq. (12):

$$\tilde{\boldsymbol{p}}^{(v_i)} = [\boldsymbol{p}^{(v_i)}; \rho]. \tag{20}$$

To incorporate virtual samples in the optimal transport problem, we construct an extended correlation matrix $\tilde{\boldsymbol{S}}$ following Chapel et al. (2020):

$$\tilde{\boldsymbol{S}} = \begin{bmatrix} \boldsymbol{S} & \boldsymbol{0}_{N \times 1} \\ \boldsymbol{0}_{1 \times N} & A \end{bmatrix}, \tag{21}$$

where $\boldsymbol{S} \in \mathbb{R}_{+}^{N \times N}$ with $\boldsymbol{S}_{ij} = s(\boldsymbol{z}_i^{(v_1)}, \boldsymbol{z}_j^{(v_2)})$, and $A < \min(\boldsymbol{S}_{ij})$. Since the objective is to maximize the expected correlation, assigning the smallest correlation value $A$ to the virtual–virtual interaction and setting all data-virtual correlation as $0$, ensures the virtual samples do not introduce a constant bias into the overall correlation score.

By adding an entropy regularization term $\mathcal{H}(\tilde{\boldsymbol{P}}) = -\sum_{i,j} \tilde{\boldsymbol{P}}_{ij} \log \tilde{\boldsymbol{P}}_{ij}$, the optimization objective can be formulated as:

$$\arg\max_{\tilde{\boldsymbol{P}} \geq 0} \langle \tilde{\boldsymbol{P}}, \tilde{\boldsymbol{S}} \rangle + \lambda \mathcal{H}(\tilde{\boldsymbol{P}}) \quad \text{s.t.} \quad \tilde{\boldsymbol{P}}\boldsymbol{1} = \tilde{\boldsymbol{p}}^{(v_1)}, \ \tilde{\boldsymbol{P}}^{\top}\boldsymbol{1} = \tilde{\boldsymbol{p}}^{(v_2)}, \tag{22}$$

where $\lambda > 0$ is a regularization factor. The augmented objective function is strictly convex and smooth. To derive the solution, we introduce the Lagrangian with dual multipliers $\boldsymbol{\alpha}, \boldsymbol{\beta} \in \mathbb{R}^{N+1}$ enforcing the row and column constraints, respectively:

$$\mathcal{L}(\tilde{\boldsymbol{P}}, \boldsymbol{\alpha}, \boldsymbol{\beta}) = \langle \tilde{\boldsymbol{P}}, \tilde{\boldsymbol{S}} \rangle - \lambda \sum_{i,j} \tilde{\boldsymbol{P}}_{ij} \log \tilde{\boldsymbol{P}}_{ij} + \boldsymbol{\alpha}^{\top}(\tilde{\boldsymbol{p}}^{(v_1)} - \tilde{\boldsymbol{P}}\boldsymbol{1}) + \boldsymbol{\beta}^{\top}(\tilde{\boldsymbol{p}}^{(v_2)} - \tilde{\boldsymbol{P}}^{\top}\boldsymbol{1}). \tag{23}$$

Taking the first-order optimality condition with respect to $\tilde{\boldsymbol{P}}_{ij}$, for any $i, j$, we have:

$$\frac{\partial \mathcal{L}}{\partial \tilde{\boldsymbol{P}}_{ij}} = \tilde{\boldsymbol{S}}_{ij} - \lambda(1 + \log \tilde{\boldsymbol{P}}_{ij}) - \boldsymbol{\alpha}_i - \boldsymbol{\beta}_j = 0. \tag{24}$$

Rearranging the terms yields:

$$\log \tilde{\boldsymbol{P}}_{ij} = \frac{\tilde{\boldsymbol{S}}_{ij} - \boldsymbol{\alpha}_i - \boldsymbol{\beta}_j}{\lambda} - 1. \tag{25}$$

By absorbing the constant terms into the scaling vectors, we obtain a multiplicative form of the solution:

$$\tilde{\boldsymbol{P}}_{ij} = \tilde{\boldsymbol{u}}_i \exp(\tilde{\boldsymbol{S}}_{ij}/\lambda) \tilde{\boldsymbol{v}}_j, \tag{26}$$

where $\tilde{\boldsymbol{u}}_i := \exp(-\boldsymbol{\alpha}_i/\lambda - 1/2)$ and $\tilde{\boldsymbol{v}}_j := \exp(-\boldsymbol{\beta}_j/\lambda - 1/2)$ are strictly positive scaling factors. In matrix form, this is expressed as:

$$\tilde{\boldsymbol{P}} = \mathrm{Diag}(\tilde{\boldsymbol{u}}) \exp(\tilde{\boldsymbol{S}}/\lambda) \mathrm{Diag}(\tilde{\boldsymbol{v}}). \tag{27}$$

Imposing the marginal constraints $\tilde{\boldsymbol{P}}\boldsymbol{1} = \tilde{\boldsymbol{p}}^{(v_1)}$ and $\tilde{\boldsymbol{P}}^{\top}\boldsymbol{1} = \tilde{\boldsymbol{p}}^{(v_2)}$ leads to the following system:

$$\mathrm{Diag}(\tilde{\boldsymbol{u}})(\exp(\tilde{\boldsymbol{S}}/\lambda)\tilde{\boldsymbol{v}}) = \tilde{\boldsymbol{p}}^{(v_1)}, \qquad \mathrm{Diag}(\tilde{\boldsymbol{v}})(\exp(\tilde{\boldsymbol{S}}^{\top}/\lambda)\tilde{\boldsymbol{u}}) = \tilde{\boldsymbol{p}}^{(v_2)}. \tag{28}$$

Solving these equations via fixed-point iteration results in the alternating Sinkhorn updates (Cuturi, 2013):

$$\tilde{\boldsymbol{u}} \leftarrow \tilde{\boldsymbol{p}}^{(v_1)} \oslash (\exp(\tilde{\boldsymbol{S}}/\lambda)\tilde{\boldsymbol{v}}), \qquad \tilde{\boldsymbol{v}} \leftarrow \tilde{\boldsymbol{p}}^{(v_2)} \oslash (\exp(\tilde{\boldsymbol{S}}^{\top}/\lambda)\tilde{\boldsymbol{u}}), \tag{29}$$

where $\oslash$ denotes element-wise division. By the Sinkhorn updates, the alternating scaling converges to unique positive vectors $(\tilde{\boldsymbol{u}}, \tilde{\boldsymbol{v}})$ that satisfy the predefined marginals. Consequently, the resulting $\tilde{\boldsymbol{P}}^*$ is the unique global maximizer of the entropy-regularized problem.

Finally, the optimal joint distribution $\boldsymbol{P}^*$ is obtained by discarding the last row and column of the augmented matrix $\tilde{\boldsymbol{P}}^*$, i.e., $\boldsymbol{P}^* = \tilde{\boldsymbol{P}}^*_{1:N,1:N}$.

# B  CONTRASTIVE LEARNING AS A SPECIAL CASE OF CORRGEN (PROOF OF PROPOSITION 2)

Starting from our generative objective in Eq. (8):

$$
\theta^* = \arg\max_{\theta} \sum_{i=1}^{N} \sum_{j=1}^{N} p(\boldsymbol{x}_j^{(v_2)}; \boldsymbol{x}_i^{(v_1)}, \theta^{(t)}) \log p(\boldsymbol{x}_i^{(v_1)}, \boldsymbol{x}_j^{(v_2)}; \theta). \tag{30}
$$

Under the assumption that the posterior collapses to $p(\boldsymbol{x}_i^{(v_2)}; \boldsymbol{x}_i^{(v_1)}, \theta) = 1$, the summation over $j$ reduces to

$$
\theta^* = \arg\max_{\theta} \sum_{i=1}^{N} \log p(\boldsymbol{x}_i^{(v_1)}, \boldsymbol{x}_i^{(v_2)}; \theta). \tag{31}
$$

Further decomposing the joint probability gives

$$
p(\boldsymbol{x}_i^{(v_1)}, \boldsymbol{x}_i^{(v_2)}; \theta) = p(\boldsymbol{x}_i^{(v_2)}; \boldsymbol{x}_i^{(v_1)}, \theta) \, p(\boldsymbol{x}_i^{(v_1)}; \theta). \tag{32}
$$

If the marginal $p(\boldsymbol{x}_i^{(v_1)}; \theta)$ is uniform, *i.e.*, $p(\boldsymbol{x}_i^{(v_1)}; \theta) = \frac{1}{N}$, it contributes only a constant independent of $\theta$, which can be omitted. Thus, the objective simplifies to

$$
\theta^* = \arg\max_{\theta} \sum_{i=1}^{N} \log p(\boldsymbol{x}_i^{(v_2)}; \boldsymbol{x}_i^{(v_1)}, \theta), \tag{33}
$$

After parameterizing the conditional probability with similarity in the embedding space, it yields exactly the InfoNCE objective (He et al., 2020):

$$
\theta^* = \arg\max_{\theta} \sum_{i=1}^{N} \log \frac{\exp(s(\boldsymbol{z}_i^{(v_1)}, \boldsymbol{z}_i^{(v_2)})/\tau)}{\sum_{n=1}^{N} \exp(s(\boldsymbol{z}_i^{(v_1)}, \boldsymbol{z}_n^{(v_2)})/\tau)}. \tag{34}
$$

# C  IMPLEMENTATION DETAILS

**Implementation of CorreGen.** CorreGen is implemented on top of DIVIDE (Lu et al., 2024). Specifically, we replace the original contrastive objective in DIVIDE with our generative objective, while retaining its feature extraction structure as the mapping function $f_\theta$. For the within-view contrastive module (*i.e.*, between features and their momentum counterparts), we fuse the estimated posterior matrix $\boldsymbol{Q}$ with the identity matrix $\boldsymbol{I}$ at a ratio of $\beta = 0.5$. For the cross-view learning module, we directly use the estimated posterior matrix without modification. To ensure stable training, we initialize the EM algorithm with the identity matrix $\boldsymbol{I}$ as the posterior estimate in the first few iterations, which serves as a warm start to avoid poor local optima. After this warmup phase, we switch to the adaptive posterior estimation strategy described in our method, thereby uncovering latent correspondences across views.

**Training Setup.** We implement CorreGen with PyTorch 2.1.2 and optimize it using the Adam optimizer (Kingma & Ba, 2014) with the learning rate of 0.002. The batch size is set to 512 for smaller datasets (*e.g.*, Scene15, LandUse21) and 1024 for larger ones (*e.g.* Caltech101, UMPC-Food101). All experiments are conducted on Ubuntu 20.04 with NVIDIA 3090 GPUs. We set the maximum warmup phase to 50 epochs and train for a total of 200 epochs. The regularization parameter $\lambda = 0.03$, and the noise ratio for the virtual sample in Eq. (12) is set to $\rho = 0.2$ across all experiments.

**Datasets.** We evaluate our method on four widely used multi-view benchmarks:

- **Scene15** (Fei-Fei & Perona, 2005) contains 4,485 natural images spanning 15 scene categories, covering both indoor and outdoor scenarios. We extract two types of hand-crafted features for each image, namely, PHOG and GIST descriptors.
- **Caltech101** (Li et al., 2015) includes 8,677 images from 101 object categories. To form two distinct views, we adopt deep representations obtained from DECAF and VGG19 networks, consistent with Han et al. (2021).

- **LandUse21** (Yang & Newsam, 2010) contains 2,100 satellite imagery samples in 21 categories. We follow Lin et al. (2022) to construct two views by extracting PHOG and LBP descriptors.

- **UMPC-Food101** (Wang et al., 2015) consists of paired food images and textual recipes, with 60,000 samples for training and 20,000 samples for testing across 101 categories. We use the test split for clustering evaluation. Visual features are extracted using a ViT (Wu et al., 2020) pretrained on ImageNet, while textual features are obtained with BERT (Devlin et al., 2018). Notably, the recipe descriptions often contain irrelevant or noisy information, making UMPC-Food101 a realistic benchmark for studying noisy correspondence.

**Simulation of sample-level mismatch.** To evaluate robustness under different conditions, we simulate two types of sample-level mismatches: i) *Alignable mismatch*: a fraction of instances (each with multiple views) are randomly permuted across views. The fraction is controlled by the *Mismatch Ratio (MR)*. ii) *Unalignable mismatch*: a fraction of view samples are corrupted with random Gaussian noise, with the fraction defined as the *Corruption Ratio (CR)*.

# D   PERFORMANCE VISUALIZATION WITH VARYING MR AND CR VALUE (Q3)

Previous comparisons in Section 4.2 focused on specific MR and CR values, which do not fully reveal robustness across different mismatch levels. Here, we fix MR at two representative values and vary CR continuously, visualizing clustering performance of CorreGen and four state-of-the-art baselines to examine their robustness.

For evaluation, we re-align samples across views using a nearest-neighbor principle following Guo et al. (2024); Sun et al. (2025). To quantify category-level consistency, we report the *Category-level Alignment Ratio* (CAR) (Yang et al., 2021), defined as

$$\text{CAR} = \frac{1}{N} \sum_{i=1}^{N} \delta \left( C(\boldsymbol{x}_i^{(v_1)}), , C(\boldsymbol{x}_{\pi(i)}^{(v_2)}) \right), \tag{35}$$

where $C(\cdot)$ is the oracle category label (Kou et al., 2025), $\pi(i)$ is the re-aligned counterpart of $\boldsymbol{x}_i^{(v_1)}$, and $\delta(\cdot)$ is the indicator function. As shown in Fig. 4, on UMPC-Food101, CorreGen demonstrates substantially lower performance degradation as CR increases, consistently outperforming all baselines. Even under severe mismatches (*e.g.*, MR=0.5), CorreGen maintains a stable CAR score, highlighting its ability to recover reliable category-level correspondences despite high noise.

# E   PARAMETERS ANALYSIS (Q4)

In this section, we provide a detailed sensitivity analysis of CorreGen using the Scene15 (Fei-Fei & Perona, 2005) dataset under the setting (MR = 0.2, CR = 0.2). We focus on three critical hyperparameters in the E-step: the pre-defined noise ratio $\rho$, the number of Sinkhorn iterations $t$, and the curve-shaping parameter $m$. To study potential interactions, we examine them in two pairwise groups.

**Pre-defined Noise Ratio $\rho$ and Curve-Shaping Parameter $m$.** As shown in Fig. 5, the performance remains stable across a wide range of $\rho$ values. For $m$, the performance is consistently strong when $m \leq 10$, where the marginal probabilities remain moderately discriminative. As $m$ grows larger, the probability distribution becomes overly smoothed, leading to a slight decline in performance.

**Pre-defined Noise Ratio $\rho$ and Sinkhorn Iterations $t$.** Fig. 6 illustrates the clustering performance of our method across a wide range of Sinkhorn iterations ($t \in [10, 1000]$) and pre-defined noise ratio ($\rho \in [0.1, 0.5]$). We observe that while increasing the number of iterations leads to a modest performance gain, the method remains comparable even with a small number of iterations. This stability is particularly advantageous as it preserves high computational efficiency without compromising accuracy. Furthermore, when $\rho$ is close to the underlying noise ratio (*e.g.*, 0.1-0.2), selecting an appropriate number of iterations enables the model to achieve optimal performance.

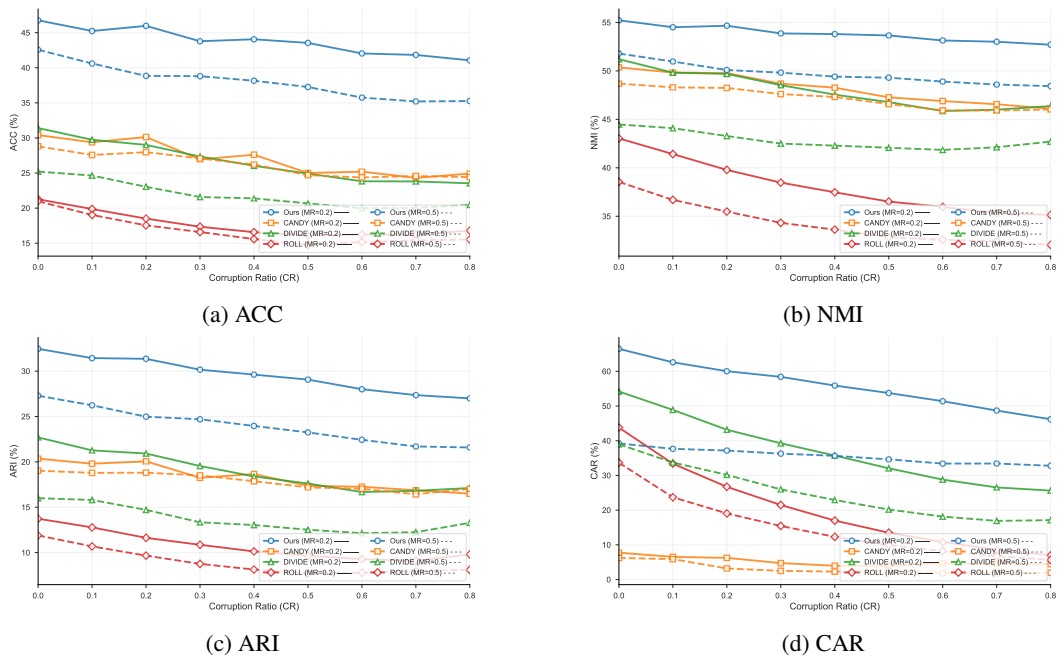

Figure 4: The clustering performance under varying CR value. Solid lines indicate results with $MR = 0.2$, while dashed lines correspond to $MR = 0.5$. The CR values varies from $0.0$ to $0.8$.

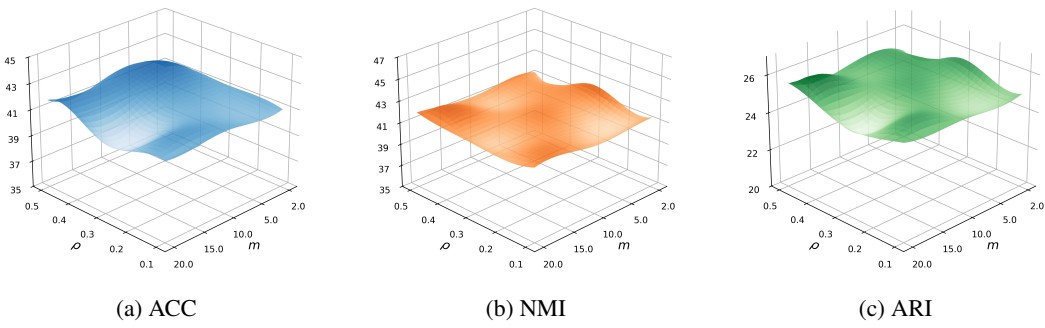

Figure 5: Parameters Analysis of Pre-defined noisy ratio $\rho$ and the curve-shaping parameter $m$.

## F  ABLATION STUDIES (Q5)

In this section, we conduct ablation studies on Scene15 and UMPC-Food101 to evaluate the effectiveness of each component. We also compare our method CorreGen with the standard In-foNCE objective. Experiments are performed under two settings: $(MR = 0.0, CR = 0.0)$ and $(MR = 0.2, CR = 0.2)$.

As shown in Table 3, the results lead to three key observations: i) On relatively clean datasets, the effect of the Virtual Sample module is not significant, and using a smaller $\rho$ may yield better results; ii) The GMM-guided marginal estimation consistently enhances clustering accuracy by assigning higher probabilities to informative samples, thereby improving joint distribution estimation. iii) Training with vanilla InfoNCE fails to capture latent sample- and category-level correspondences, resulting in significant performance degradation under noisy conditions.

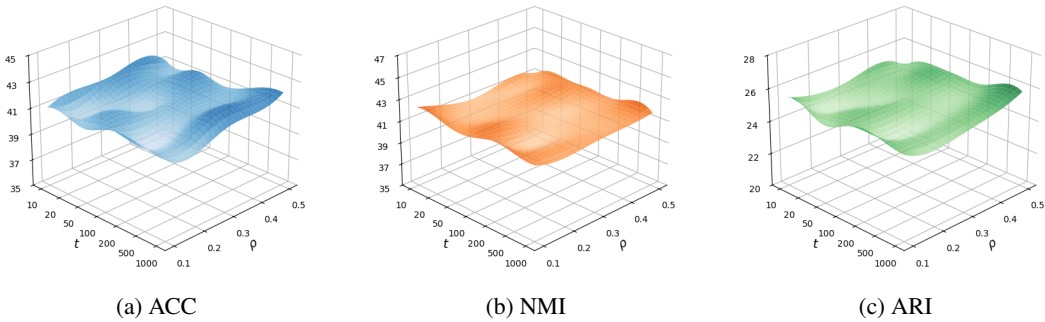

| (a) ACC | (b) NMI | (c) ARI |

Figure 6: Parameters Analysis of pre-defined noise ratio $\rho$ and Sinkhorn iterations $t$.

Table 3: Ablation study of CorreGen on Scene15 and UMPC-Food101, where *w/o* denotes the component is not adopted. "Virtual" refers to the Virtual Sample module, "Guide" refers to the GMM-guided marginal estimation, and "Vanilla InfoNCE" denotes training with the standard contrastive objective.

| Setting | MR=0.0, CR=0.0 | | | | | | MR=0.2, CR=0.2 | | | | | |
| | Scene15 | | | UMPC-Food101 | | | Scene15 | | | UMPC-Food101 | | |
| | ACC | NMI | ARI | ACC | NMI | ARI | ACC | NMI | ARI | ACC | NMI | ARI |
| **CorreGen** | **50.25** | **48.92** | **32.87** | **49.77** | 58.36 | 35.73 | **41.78** | **41.67** | **25.50** | **45.97** | **54.66** | **31.36** |
| *w/o* Virtual | 49.44 | 48.38 | 32.15 | 49.45 | **59.22** | **36.65** | 41.10 | 41.12 | 24.77 | 44.01 | 53.92 | 30.36 |
| *w/o* Guide | 49.06 | 48.01 | 31.98 | 49.44 | 57.95 | 35.37 | 40.98 | 41.21 | 24.77 | 44.59 | 54.03 | 30.67 |
| *w/o* Virtual & Guide | 49.00 | 48.33 | 31.83 | 48.92 | 58.42 | 35.61 | 40.52 | 40.95 | 24.66 | 43.68 | 53.41 | 29.78 |
| Vanilla InfoNCE | 47.83 | 47.81 | 31.37 | 48.47 | 57.82 | 34.73 | 38.36 | 37.60 | 21.96 | 43.84 | 52.76 | 29.15 |

## G  CONVERGENCE ANALYSIS

In this section, we analyze the convergence behavior of CorreGen from both theoretical and empirical perspectives to demonstrate the training stability of our proposed framework.

**Theoretical Analysis**. In the two-view case, our optimization objective is the likelihood function:

$$\mathcal{L}(\theta) = \sum_{i=1}^{N} \log \sum_{j=1}^{N} p(x_i^{(v_1)}, x_j^{(v_2)}; \theta). \tag{36}$$

By introducing an auxiliary distribution $Q(x_j^{(v_2)})$ for each sample $x_i^{(v_1)}$, we derive a lower bound via Jensen's inequality:

$$\mathcal{L}(\theta) \geq \sum_{i=1}^{N} \sum_{j=1}^{N} Q(x_j^{(v_2)}) \log \frac{p(x_i^{(v_1)}, x_j^{(v_2)}; \theta)}{Q(x_j^{(v_2)})} \triangleq \mathcal{B}(Q, \theta), \tag{37}$$

which holds with equality when $Q(x_j^{(v_2)}) = p(x_j^{(v_2)}; x_i^{(v_1)}; \theta)$. In the E-step, we estimate $Q^{(t+1)}(x_j^{(v_2)}) = p(x_j^{(v_2)}; x_i^{(v_1)}; \theta^{(t)})$ to make the bound tight such that

$$\mathcal{L}(\theta^{(t)}) = \mathcal{B}(Q^{(t+1)}, \theta^{(t)}). \tag{38}$$

Subsequently, the M-step updates $\theta$ to maximize this expected log-likelihood, ensuring $\mathcal{B}(Q^{(t+1)}, \theta^{(t+1)}) \geq \mathcal{B}(Q^{(t+1)}, \theta^{(t)})$. Combining these steps yields the following inequality chain, proving that the likelihood is monotonically non-decreasing:

$$\mathcal{L}(\theta^{(t+1)}) \geq \mathcal{B}(Q^{(t+1)}, \theta^{(t+1)}) \geq \mathcal{B}(Q^{(t+1)}, \theta^{(t)}) = \mathcal{L}(\theta^{(t)}). \tag{39}$$

Given that the likelihood function is bounded, this monotonicity guarantees the convergence of our algorithm to a stationary point.

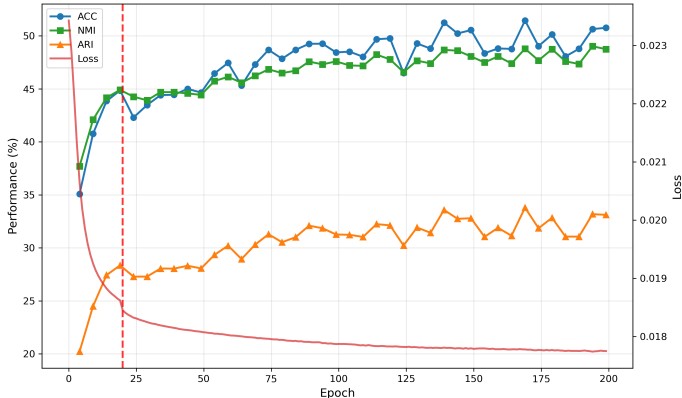

Figure 7: Convergence analysis of CorreGen on the Scene15 dataset. The red dashed line indicates the transition from the warmup phase to the EM optimization phase.

**Empirical Verification.** To empirically verify this stability, we tracked the training loss and clustering performance on the Scene15 dataset over the training process. As our optimization performs gradient descent on the *negative* expected log-likelihood, the loss naturally decreases as training progresses. As illustrated in Fig. 7, the training process exhibits a clear and stable convergence pattern. During the initial warm-up phase (epochs 0–20), the loss decreases rapidly while performance metrics show a sharp increase. After the EM procedure is activated (marked by the red dashed line), the loss continues to decline steadily. After approximately 150 epochs, both the objective function and all evaluation metrics stabilize and reach a plateau, confirming that our objective has converged.

## H   ANALYSIS OF CATEGORY-LEVEL MISMATCH RATIO

In this section, we first provide the mathematical formulation of the Category-level Mismatch Ratio (CMR). For a category $c$ containing $N_c$ samples, the total space of possible cross-view pairwise interactions in this category is $N_c^2$. According to Definition 1, the mismatch ratio $\gamma_c$ for category $c$ is calculated as:

$$\gamma_c = 1 - \frac{\sum_{i,j \in \text{Category } c} \mathbb{I}(t_{i,j} = 1)}{N_c^2} = 1 - \frac{N_c}{N_c^2} = 1 - \frac{1}{N_c}, \tag{40}$$

where $t_{ij}$ denotes whether the pair $(i, j)$ is an observed correspondence as defined in Definition 1, and $\mathbb{I}(t_{i,j} = 1) = N_c$ because each sample has exactly one observed correspondence in existing datasets.

Specifically, for a dataset with $C$ categories, CMR is defined as the average across categories:

$$\text{CMR} = \frac{1}{C} \sum_{i=1}^{C} \gamma_i = 1 - \frac{1}{C} \sum_{i=1}^{C} \frac{1}{N_i}. \tag{41}$$

According to the above formulation, Table 4 reports the CMR of datasets used in our experiments. The results indicate that category-level mismatches are pervasive across datasets (consistently exceeding 98%), highlighting the necessity of uncovering latent correspondences beyond the limited off-the-shelf pairs.

Furthermore, we analyze the behavior of this metric under a fixed data size $N$, *i.e.*, $\sum_{i=1}^{C} N_i = N$. As Eq. (40) is strictly concave on $(0, \infty)$, applying Jensen's Inequality for concave functions yields:

$$\frac{1}{C} \sum_{i=1}^{C} \left(1 - \frac{1}{N_i}\right) \le 1 - \frac{1}{\frac{1}{C} \sum_{i=1}^{C} N_i} = 1 - \frac{C}{N}. \tag{42}$$

The equality holds if and only if $N_1 = N_2 = \cdots = N_C$, when the dataset is perfectly balanced. This inequality implies that balanced datasets will inherently exhibit a higher average category-level mismatch ratio compared to imbalanced or long-tailed datasets of the same size.

Table 4: The Category-level Mismatch Ratio (CMR) for the datasets used in our experiments.

| Dataset | CMR (%) |
|---|---|
| Scene15 | 99.65 |
| LandUse21 | 99.00 |
| Caltech101 | 98.25 |
| UMPC-Food101 | 99.53 |

| Text | Image | Label |
|---|---|---|
| The unincorporated community of Pie Town, New Mexico is named in honour of the apple pie. [ 20 ]
## See also [ edit ]
Food portal
 * Apfelstrudel ( apple strudel ), an Austrian pie-like dish made with dough.
 * Apple cake
 * Apple cobbler
 * Tarte Tatin , a French variant on apple pie.
## References [ edit ] |  | Apple Pie |
| Why oh why did I not think of these? I love Chicken Pot Pie, but have never ever made it. Weird, I know. I don't know why I haven.
 * BBQ Pulled Pork - Slow Cooker Thursday
Hello everyone! To all of my American friends, Happy Thanksgiving to you! Here in Canada we celebrate Thanksgiving in October, so for us...
 * Blueberry Maple Muffins
You've heard me say this before, but I love muffins. They are such an easy thing to eat when you've got a baby attached to your hi...
Watermark template. Powered by Blogger .
9:06 PM
]: 2012-01-27T21:06:00-08:00 |  | baby back ribs |
| Latest Competitions
Kambrook Express Digital Pressure Cooker
 * Kambrook Express Digital Pressure Cooker
 * Kenwood kMix Food Processor
 * Messermeister Oliva Elité knives
 * Win a trip to Italy!
 * Win a family trip to the Gold Coast with Rio 2
 * Win a trip to Croatia
 * Magazine * Meal Plans * Experts
 * Features * Social * Subscribe |  | beef carpaccio |
| ##### Good Food Good Times Series 1
In this series Barry Lewis from YouTube channel My Virgin Kitchen shows you how to make some great recipes for the whole family - including naan bread pizzas, spicy salmon burgers with pineapple and quinoa chilli chow!
#### Chef: The Videojug Team
### You May Like
 * _ _
##### Cook Beef Tenderloin |  | beignets |
| ** Links: **
Ruhlman's chicken-fried pork belly ceasar
Filipino Pantry Chicken Caesar Salad from Burnt Lumpia
Chicken Caesar Salad made with a buttermilk dressing, from Cafe Fernando
Caesar Salad Club Sandwich from Noble Pig
Caesar Salad with Shrimp from Lydia of The Perfect Pantry
Share on FacebookTweet34 |
Filed under Crouton , Garlic , Romaine Lettuce , Salad
### Never miss a recipe! (details) |  | caprese_salad |

Figure 8: Examples of noisy image-text pair in the UMPC-Food101 dataset.

## I   IMAGE-TEXT PAIR EXAMPLE OF UMPC-FOOD101

UMPC-Food101 (Wang et al., 2015) is constructed by crawling food images with textual recipes collected from the web. As shown in Fig. 8, the texts often contain irrelevant descriptions, hyperlinks, or noisy information unrelated to the visual content, making it a realistic benchmark for studying noisy correspondence in multi-view clustering.

## J   THE USE OF LARGE LANGUAGE MODELS

In this paper, LLMs were used to refine the writing in the Introduction, Related Work, and Experiments sections, as well as to verify the clarity of mathematical derivations.

