# OpenReview forum: "Uncover Underlying Correspondence for Robust Multi-view Clustering"
_ICLR.cc/2026/Conference — ICLR 2026 Oral_

### Official Review · Reviewer_YFa9 · 2025-10-30

**Soundness:** 3
**Presentation:** 3
**Contribution:** 4
**Rating:** 8
**Confidence:** 5

**Summary:**

This paper addresses noisy correspondence in real-world multi-view clustering. The authors identify category-level mismatch and sample-level mismatch. The paper proposes a generative framework based on maximum likelihood estimation and optimized via an Expectation-Maximization algorithm. The framework infers a soft correspondence distribution in the E-step (guided by GMM-based marginals and optimal transport) and refines embeddings in the M-step. Experiments on four benchmark datasets demonstrate superior robustness under various noisy settings, including up to 80% mismatch rates.

**Strengths:**

1）The paper presents a thoughtful formulation by explicitly distinguishing two realistic types of noisy correspondence in multi-view data, namely, category-level mismatch and sample-level mismatch, providing a solid rationale for why existing contrastive MVC methods fall short under such conditions.
2）Instead of relying on pairwise contrastive learning, the authors shift to a generative modeling perspective, leveraging an Expectation-Maximization algorithm to uncover latent correspondences. This allows them to model complex many-to-many, probabilistic associations between samples across views.
3）The method demonstrates consistently strong performance across diverse datasets and under various levels of noise and corruption.

**Weaknesses:**

1）The related work section does not adequately cover more recent progress, especially in areas concerning missing modalities and partially view-aligned representation learning.
2）Although warm-start and momentum updates are mentioned as strategies to stabilize training, the paper does not provide convergence curve analyses.
3）The paper does not include comparisons with recent methods that also employ optimal transport for multi-view clustering. Incorporating an experimental comparison with approaches such as “PROTOCOL: Partial Optimal Transport-enhanced Contrastive Learning for Imbalanced Multi-view Clustering” would strengthen the work.

**Questions:**

1）Could the authors provide a more comprehensive comparison with recent advances on incomplete multi-view and PVP problems, especially those published post 2022?
2）Can the authors provide quantitative or visual evidence (e.g., convergence plots) to support the stability and convergence behavior of CorreGen’s EM algorithm?
3）How does CorreGen compare with other OT-based multi-view methods in terms of robustness?

---

> ### Author Response · Authors · 2025-11-25
>
> We thank the reviewer for the positive assessment and valuable suggestions. In the revised manuscript, we have updated the related work with recent methods, added a formal convergence analysis, and included a comparative evaluation against OT-based methods. We address the specific questions as follows.
>
> > **Q1: Comparison in Related Work**
>
> We sincerely thank the reviewer for this helpful suggestion. In the revised manuscript, we have expanded the *Related Work* section to include more recent advances in both Incomplete Multi-view Learning (IMP) and the Partially View-aligned Problem (PVP).
>
> For IMP, we have incorporated cutting-edge approaches including anchor learning [A], tensorized subspace learning [B], and diffusion-based generation [C]. For PVP, we have added discussions on recent techniques such as the multi-stage re-alignment strategy [D] and robust variational contrastive learning [E].
>
> [A] Liu S, Wang S, Liang K, et al. Alleviate anchor-shift: Explore blind spots with cross-view reconstruction for incomplete multi-view clustering[J]. Advances in Neural Information Processing Systems, 2024, 37: 87509-87531.
>
> [B] Zhang G Y, Huang D, Wang C D. Unified and tensorized incomplete multi-view kernel subspace clustering[J]. IEEE Transactions on Emerging Topics in Computational Intelligence, 2024, 8(2): 1550-1566.
>
> [C] Zhang Y, Lin Y, Yan W, et al. Incomplete Multi-view Clustering via Diffusion Contrastive Generation[C]//Proceedings of the AAAI Conference on Artificial Intelligence. 2025, 39(21): 22650-22658.
>
> [D] Yan W, Zhu J, Chen J, et al. Partially multi-view clustering via re-alignment[J]. Neural Networks, 2025, 182: 106884.
>
> [E] He C, Zhu H, Hu P, et al. Robust variational contrastive learning for partially view-unaligned clustering[C]//Proceedings of the 32nd ACM International Conference on Multimedia. 2024: 4167-4176.
>
> > **Q2: Convergence analysis**
>
> We thank the reviewer for the insightful suggestion. In the revised manuscript, we have added a formal convergence analysis of our EM framework in **Appendix G**. In addition, we report the training loss curve on Scene15 (see **Figure 7**). The loss decreases rapidly in the early stage and stabilizes after roughly 25 epochs, empirically demonstrating the stability and convergence behavior of our method.
>
> > **Q3: Comparison to other OT-based multi-view learning methods**
>
> Thanks for pointing out this important baseline. We evaluated [**PROTOCOL**](https://arxiv.org/pdf/2506.12408) following exactly the same experimental settings as in our paper. As shown below, CorreGen consistently outperforms PROTOCOL across different datasets.
>
> | Method   | Setting (MR / CR) | Scene15 (ACC \| NMI \| ARI) | Caltech101 (ACC \| NMI \| ARI) | LanUse21 (ACC \| NMI \| ARI) | UMPC_Food101 (ACC \| NMI \| ARI) |
> | -------- | :---------------: | :-------------------------: | :----------------------------: | ---------------------------- | -------------------------------- |
> | PROTOCOL |     0.0 / 0.0     |   42.65 \| 42.98 \| 24.87   |    56.00 \| 81.00 \| 39.00     | 24.62 \| 28.72 \| 11.05      | 11.66 \| 22.17 \| 3.39           |
> | Ours     |     0.0 / 0.0     |    50.25 \| 48.92 \| 32.87    |    68.52 \| 84.45 \| 63.45     | 32.87 \| 39.52 \| 18.54      | 49.77 \| 58.36 \| 35.73          |
> | PROTOCOL |     0.2 / 0.0     |   36.61 \| 30.38 \| 17.44   |    48.14 \| 69.38 \| 28.65     | 25.38 \| 27.23 \| 11.38      | 9.79 \| 16.93 \| 2.17            |
> | Ours     |     0.2 / 0.0     |   48.04 \| 47.36 \| 30.75   |    68.01 \| 84.23 \| 62.78     | 32.26 \| 38.76 \| 17.83      | 46.76 \| 55.22 \| 32.46          |
> | PROTOCOL |     0.5 / 0.0     |   36.50 \| 28.82 \| 17.01   |    45.51 \| 66.52 \| 29.93     | 26.90 \| 28.08 \| 12.21      | 5.89 \| 9.99 \| 0.66             |
> | Ours     |     0.5 / 0.0     |   45.07 \| 44.97 \| 27.87   |    66.60 \| 83.61 \| 62.38     | 32.03 \| 37.98 \| 17.84      | 42.57 \| 51.79 \| 27.29          |
> | PROTOCOL |     0.2 / 0.2     |   32.46 \| 28.11 \| 15.05   |    49.18 \| 72.47 \| 31.25     | 23.33 \| 24.65 \| 9.57       | 8.29 \| 15.09 \| 1.72            |
> | Ours     |     0.2 / 0.2     |   41.23 \| 41.43 \| 25.05   |    67.12 \| 84.45 \| 64.13     | 31.13 \| 37.36 \| 17.00      | 45.97 \| 54.66 \| 31.36          |
> | PROTOCOL |     0.5 / 0.2     |   24.48 \| 15.23 \| 8.00    |    44.11 \| 68.22 \| 30.39     | 24.90 \| 23.95 \| 9.81       | 5.26 \| 8.85 \|0.51              |
> | Ours     |     0.5 / 0.2     |   39.54 \| 39.55 \| 23.12   |    66.87 \| 84.15 \| 67.31     | 31.20 \| 36.25 \| 16.92      | 38.84 \| 50.05 \| 24.98          |

---

### Official Review · Reviewer_iE2N · 2025-10-30

**Soundness:** 3
**Presentation:** 3
**Contribution:** 3
**Rating:** 6
**Confidence:** 4

**Summary:**

The paper addresses the challenge of noisy correspondence (NC) in multi-view clustering (MVC), which arises when data collected from the web contains misaligned or unmatchable samples. The authors distinguish two harmful types of NC: 1) Category-level mismatch: samples from the same class treated as negative; 2) Sample-level mismatch: mispaired or unalignable samples. To tackle this, they propose CorreGen, a generative EM-based framework that formulates correspondence discovery as maximum likelihood estimation. In the E-step, they estimate soft correspondence distributions using optimal transport constrained by GMM-guided marginals and augmented with virtual samples to absorb outliers. In the M-step, they update embeddings to maximize expected log-likelihood using these inferred correspondences. Extensive experiments on synthetic and real datasets show strong robustness with substantial improvements under both category- and sample-level mismatches.

**Strengths:**

1. This paper shifts from discriminative pairwise reweighting/realignment to a principled maximum likelihood framework, reducing reliance on noisy pre-defined pairs.
2. The proposed method consistently outperforms state-of-the-art baselines under varying mismatch and corruption rates, especially on the noisy UMPC-Food101 dataset.

**Weaknesses:**

1. While the EM-based optimal transport is elegant, they may introduce computational overhead when applied to very large-scale or high-dimensional datasets. The paper would benefit from a more detailed runtime analysis.
2. The paper should clarify why the posterior weight $Q_{ij}$ is defined as $\frac{\boldsymbol{P}_{ij}^*}{\boldsymbol{p}_i^{(v_1)}}$ in Equation (17).
3. In Equation (17), the matrix ${Q}$ appears to serve as a re-alignment signal that guides the model's learning. It would be informative to analyze what happens if ${Q}$ is replaced with an identity matrix, and how would performance change if ${Q}$ were substituted with the ground-truth correspondence matrix?
4. A minor presentation issue: on page 2, the text refers to Figure 2, but it seems this should be Figure 1.

**Questions:**

Please refers to the weaknesses.

---

> ### Author Response · Authors · 2025-11-25
>
> We are grateful for the reviewer's valuable suggestions to enhance the clarity and rigor of our manuscript. Following your suggestions, we have included a detailed computational cost analysis, refined the definition of the posterior, and conducted additional studies on the posterior. Specific responses to each question are provided below.
>
> > **Q1:Efficiency Concerns**
>
> Thank you for the valuable question regarding computational efficiency.  Below, we provide both theoretical complexity analysis and empirical runtime measurements of the EM procedure.
>
> Theoretically, the extra cost introduced by CorreGen mainly arises from two components in the E-step: **Sinkhorn iterations** and **GMM fitting**.
>
> - **Sinkhorn iteration**: Since the Sinkhorn iteration operates on mini-batches of fixed size $B$, its time complexity per epoch is $O(\frac{L N B}{\lambda^2})$, where $L$ is the number of Sinkhorn iterations, $N$ is the total sample size and $\lambda$ is a regularization term. The complexity grows **linearly** with the dataset size $N$. Characterized by dense matrix multiplications and element-wise operations, Sinkhorn is parallelizable and could be efficiently accelerated on GPUs.
> - **GMM**: The complexity is $O(I N  C D)$, where $I$, $C$, and $D$ denote iterations, number of clusters, and feature dimensions, respectively. GMM also scales **linearly** with the dataset size. In our experiments, to avoid numerical instability when processing high-dimensional embeddings, we implement GMM on the CPU, which introduces moderate overhead and could be accelerated by GPUs in the future.
>
> Beyond the theoretical analysis, we measured the training time and memory usage on three datasets with varying sizes. The results on the original datasets below show that although CorreGen increases training time mainly due to CPU-based GMM, GPU memory remains almost identical to DIVIDE, and overall computation stays within reasonable bounds.
>
> | Ours                            |        Scene15         |       Caltech101        |      UMPC_Food101       |
> | ------------------------------- | :--------------------: | :---------------------: | :---------------------: |
> | Size/ Batch size                |       4485/ 512        |        8677/1024        |       21695/1024        |
> | Avg epoch time                  |        1.4392s         |         5.0956s         |        13.8309s         |
> | Avg Sinkhorn iteration time     |        0.6286s         |         0.5715s         |         1.2197s         |
> | Avg GMM time                    |        0.5117s         |         3.8633s         |        10.7913s         |
> | GPU Memory (MB)                 |          524M          |          782M           |          836M           |
> | Performance (ACC \| NMI \| ARI) | 50.25 \| 48.92 \|32.87 | 68.52 \| 84.45 \| 63.45 | 49.77 \| 58.36 \| 35.73 |
>
> | DIVIDE                          |         Sceen15         |       Caltech101        |      UMPC_Food101       |
> | ------------------------------- | :---------------------: | :---------------------: | :---------------------: |
> | Size/Batch size                 |        4485/512         |        8677/1024        |       21695/1024        |
> | Avg epoch time                  |         0.2681s         |         0.6356s         |         1.4503s         |
> | GPU Memory (MB)                 |          518M           |          768M           |          834M           |
> | Performance (ACC \| NMI \| ARI) | 44.57 \| 45.98 \| 28.43 | 62.20 \| 83.30 \| 50.50 | 36.20 \| 57.92 \| 27.72 |
>
> For very large-scale datasets where GMM fitting may become a bottleneck, **CorreGen remains flexible**. As shown in our ablation study (**Table 3** in the manuscript), the E-step can be simplified by assuming a uniform marginal distribution and removing the GMM guidance.  Even under this simplified setting, the method still **significantly outperforms standard contrastive baselines**, showing that our framework scales well without compromising robustness.

---

> > ### Author Response · Authors · 2025-11-25
> >
> > > **Q2: The defintion of posterior weight**
> >
> > Thank you for the suggestion. We have updated the manuscript to provide a clearer and more rigorous explanation of how the posterior $Q_{ij}$ is derived in the M-step (see **Section 3.2.2** of the revised version):
> >
> > > According to Eq. (9), we compute the posterior using the optimal joint distribution ${P}^{\*}$ and marginals $p^{(v_1)}$ obtained in the E-step, defined as $Q_{ij} = P_{ij}^*/p_i^{(v_1)}$.
> >
> > This revision explicitly clarifies that $Q_{ij}$ is defined as the posterior probability of $x_j^{(v_2)}$ given $x_i^{(v_1)}$ under the current parameters:
> > $$
> > Q_{ij} \coloneqq p\big(x_j^{(v_2)}, x_i^{(v_1)}; \theta^{(t)}\big).
> > $$
> >  Specifically, the posterior is derived from the marginal distribution $p_i^{(v_1)}=p(x_i^{(v_1)};\theta^{(t)})$ and the joint distribution $P_{ij}^* = p\big(x_i^{(v_1)}, x_j^{(v_2)}; \theta^{(t)}\big)$ estimated in E-step:
> > $$
> > Q_{ij}= p\big(x_j^{(v_2)} ; x_i^{(v_1)}; \theta^{(t)}\big)= \frac{p\big(x_i^{(v_1)}, x_j^{(v_2)}; \theta^{(t)}\big)}       {p\big(x_i^{(v_1)}; \theta^{(t)}\big)}= \frac{P_{ij}^*}{p_i^{(v_1)}},
> > $$
> >
> > > **Q3: The analysis of posterior probability $Q$ be replaced by the identity matrix or groundtruth**
> >
> > Thank you for this valuable suggestion. Replacing the posterior matrix $Q$ with the **Identity matrix (**I**)** and the **Ground-Truth correspondence matrix (**GT**)** indeed provides meaningful lower and upper bounds for evaluating our framework.
> >
> > We conducted this experiment on the Scene15 and UMPC-Food101 datasets under the clean setting ($MR=0.0$, $CR=0.0$). The results are summarized below.
> >
> > | Setting              | Scene15 (ACC \| NMI \| ARI) | UMPC_Food101 (ACC \| NMI \| ARI) |
> > | :------------------- | :-------------------------: | :------------------------------: |
> > | Vanilla InfoNCE      |   47.83 \| 47.81 \| 31.37   |     48.47 \| 57.82 \| 34.73      |
> > | Identity ($Q=I$)     |   47.64 \| 47.79 \| 31.49   |     48.52 \| 57.63 \| 34.96      |
> > | GroundTruth ($Q=GT$) |   97.33 \| 97.76 \| 96.41   |     86.76 \| 97.04 \| 83.47      |
> > | CorreGen ($Q_{est}$) |   50.25 \| 48.92 \| 32.87   |     49.77 \| 58.36 \| 35.73      |
> >
> > **Lower bound (Identity matrix).**
> >
> > Setting $Q = I$ satisfies part of the degeneracy conditions discussed in **Proposition 2** of our paper, reducing our objective to a similar form of the vanilla InfoNCE loss. Empirically, the results align closely with the "Vanilla InfoNCE" objective, confirming that this substitution corresponds to a discriminative contrastive objective without correspondence discovery.
> >
> > **Upper bound (Ground-Truth matrix).**
> >
> > Using the ground-truth correspondence matrix actually replaces the unsupervised objective $Q$ with **fully supervised contrastive learning**. Since clustering typically trains and evaluates on the entire dataset without a train/test split, providing ground-truth correspondence effectively leaks full category information during training. This “oracle” setting naturally reaches near-perfect performance and reflects the upper capacity. The substantial performance gap between this oracle and our unsupervised approach is therefore expected and appropriate.
> >
> > >  **Q4: Typo**
> >
> > Thanks. We have corrected the typo in the revised manuscript.

---

### Official Review · Reviewer_jwwK · 2025-10-31

**Soundness:** 3
**Presentation:** 3
**Contribution:** 3
**Rating:** 6
**Confidence:** 4

**Summary:**

Multi-view clustering (MVC) leverages cross-view consistency but suffers from noisy correspondence (NC), including category-level mismatch and sample-level mismatch. This paper proposes a generative framework, CorreGen, which formulates NC learning as maximum likelihood estimation of underlying cross-view correspondences, solved via an Expectation-Maximization (EM) algorithm. In the E-step, soft correspondence distributions are inferred using GMM-guided marginals to down-weight noisy samples; in the M-step, the embedding network is updated to maximize expected log-likelihood. Extensive experiments on synthetic and real-world datasets demonstrate superior robustness.

**Strengths:**

1.	This paper introduces a generative framework that reduces reliance on noisy predefined pairs by modeling latent correspondences, enabling more robust semantic structure learning.
2.	This method effectively handles both category-level and sample-level mismatch via GMM-guided marginals and virtual samples, outperforming baselines on complex datasets.
3.	Ablation studies show stable performance across varying curve-shaping parameters and noise rates, reducing hyperparameter tuning burden.

**Weaknesses:**

1.	This method may incur higher computational cost due to EM iterations, GMM fitting, and optimal transport computations, potentially limiting scalability to very large datasets.
2.	This method relies on GMM assumptions, which may fail to model non-Gaussian embedding distributions, leading to suboptimal marginal estimates in complex data.
3.	The experiments focus on two-view settings, limiting generalization to multi-view scenarios where correspondence inference becomes more complex.

**Questions:**

Please refer to the Weaknesses.

---

> ### Author Response · Authors · 2025-11-25
>
> We appreciate the reviewer’s insightful comments regarding the method's efficiency and underlying assumptions. To address these points, we have provided a comprehensive complexity analysis, clarified the flexibility of our mixture model framework, and extended our evaluation to multi-view settings. Our detailed responses follow.
>
> > **Q1: Efficiency Concerns**
>
> Thank you for the valuable question regarding computational efficiency.  Below, we provide both theoretical complexity analysis and empirical runtime measurements of the EM procedure.
>
> Theoretically, the extra cost introduced by CorreGen mainly arises from two components in the E-step: **Sinkhorn iterations** and **GMM fitting**.
>
> - **Sinkhorn iteration**: Since the Sinkhorn iteration operates on mini-batches of fixed size $B$, its time complexity per epoch is $O(\frac{L N B}{\lambda^2})$, where $L$ is the number of Sinkhorn iterations, $N$ is the total sample size and $\lambda$ is a regularization term. The complexity grows **linearly** with the dataset size $N$. Characterized by dense matrix multiplications and element-wise operations, Sinkhorn is parallelizable and could be efficiently accelerated on GPUs.
> - **GMM**: The complexity is $O(I N  C D)$, where $I$, $C$, and $D$ denote iterations, number of clusters, and feature dimensions, respectively. GMM also scales **linearly** with the dataset size. In our experiments, to avoid numerical instability when processing high-dimensional embeddings, we implement GMM on the CPU, which introduces moderate overhead and could be accelerated by GPUs in the future.
>
> Beyond the theoretical analysis, we measured the training time and memory usage on three datasets with varying sizes. The results on the original datasets below show that although CorreGen increases training time mainly due to CPU-based GMM, GPU memory remains almost identical to DIVIDE, and overall computation stays within reasonable bounds.
>
> | Ours                            |        Scene15         |       Caltech101        |      UMPC_Food101       |
> | ------------------------------- | :--------------------: | :---------------------: | :---------------------: |
> | Size/ Batch size                |       4485/ 512        |        8677/1024        |       21695/1024        |
> | Avg epoch time                  |        1.4392s         |         5.0956s         |        13.8309s         |
> | Avg Sinkhorn iteration time     |        0.6286s         |         0.5715s         |         1.2197s         |
> | Avg GMM time                    |        0.5117s         |         3.8633s         |        10.7913s         |
> | GPU Memory (MB)                 |          524M          |          782M           |          836M           |
> | Performance (ACC \| NMI \| ARI) | 50.25 \| 48.92 \|32.87 | 68.52 \| 84.45 \| 63.45 | 49.77 \| 58.36 \| 35.73 |
>
> | DIVIDE                          |         Sceen15         |       Caltech101        |      UMPC_Food101       |
> | ------------------------------- | :---------------------: | :---------------------: | :---------------------: |
> | Size/Batch size                 |        4485/512         |        8677/1024        |       21695/1024        |
> | Avg epoch time                  |         0.2681s         |         0.6356s         |         1.4503s         |
> | GPU Memory (MB)                 |          518M           |          768M           |          834M           |
> | Performance (ACC \| NMI \| ARI) | 44.57 \| 45.98 \| 28.43 | 62.20 \| 83.30 \| 50.50 | 36.20 \| 57.92 \| 27.72 |
>
> For very large-scale datasets where GMM fitting may become a bottleneck, **CorreGen remains flexible**. As shown in our ablation study (**Table 3** in the manuscript), the E-step can be simplified by assuming a uniform marginal distribution and removing the GMM guidance.  Even under this simplified setting, the method still **significantly outperforms standard contrastive baselines**, showing that our framework scales well without compromising robustness.
>
> > **Q2:  This method relies on GMM assumptions**
>
> Thank you for pointing out these limitations. We fully agree that the true embedding distribution may not strictly follow a Gaussian mixture. In our framework, this choice is made for simplicity rather than being fundamental to the framework.
>
> Crucially, **CorreGen is not limited to Gaussian mixtures**. The E-step only requires a model capable of producing clustering assignments from which the mean and covariance of each component can be computed. This means CorreGen can be instantiated with general mixture models [A], including heterogeneous component families (e.g., Gaussian and Laplace) or even non-parametric mixtures, whenever prior knowledge suggests a better choice. We also note that, in the absence of such prior knowledge, Gaussian mixtures are widely used as highly expressive universal models.

---

> > ### Author Response · Authors · 2025-11-25
> >
> > > **Q3: Multi-view settings**
> >
> > Thank you for the insightful comment. Here, we extended the experiments to the 3-view setting on the Scene15 dataset. The results below show that CorreGen effectively aggregates information from multiple views.
> >
> > |  **MR / CR**  | **3-View (ACC \| NMI \| ARI)** | **2-View (ACC \| NMI \| ARI)** |
> > | :-----------: | :----------------------------: | :----------------------------: |
> > | **0.0 / 0.0** |    53.71 \| 51.98 \| 37.71     |    50.25 \| 48.92 \| 32.87     |
> > | **0.2 / 0.0** |    51.36 \| 48.99 \| 33.95     |    48.04 \| 47.36 \| 30.75     |
> > | **0.5 / 0.0** |    45.71 \| 44.24 \| 28.54     |    45.07 \| 44.97 \| 27.87     |
> > | **0.8 / 0.0** |    40.11 \| 39.14 \| 22.96     |    40.96 \| 41.74 \| 24.74     |
> > | **0.2 / 0.2** |    42.21 \| 42.16 \| 25.88     |    41.23 \| 41.43 \| 25.05     |
> > | **0.5 / 0.5** |    35.78 \| 34.79 \| 19.49     |    36.19 \| 36.84 \| 20.83     |
> >
> > In clean or mildly noisy settings (e.g., MR=0.0 or MR=0.2), the 3-view variant clearly outperforms the 2-view one, demonstrating the benefit of integrating more views.
> >
> > However, when the mismatch rate becomes extremely high (e.g., MR=0.8), the 3-view model experiences a slight performance drop compared to the 2-view model. This is expected because, in our experimental setup, **each additional view introduces an extra set of shuffled samples**, causing the total number of mismatched pairs to grow with the number of views. Consequently, the correspondence inference problem becomes more challenging as the number of views increases.
> >
> >
> >
> > [A] Pal S, Heumann C. Flexible Multivariate Mixture Models: A Comprehensive Approach for Modeling Mixtures of Non‐Identical Distributions[J]. International Statistical Review, 2024.

---

### Official Review · Reviewer_eYLV · 2025-11-01

**Soundness:** 4
**Presentation:** 3
**Contribution:** 3
**Rating:** 8
**Confidence:** 4

**Summary:**

This paper tackles noisy correspondence in multi-view clustering, arguing that real-world, web-crawled pairs contain (i) category-level mismatches (same class but treated as negatives) and (ii) sample-level mismatches including both alignable mispairs and truly unalignable samples. It proposes a generative objective that maximizes the (marginal) joint likelihood over latent cross-view correspondences and optimizes it with an EM procedure: the E-step estimates a soft, many-to-many coupling via entropy-regularized optimal transport subject to GMM-guided marginals and augmented with a virtual node to absorb outliers; the M-step maximizes a weighted log-likelihood implemented with similarity-normalized scores, unifying InfoNCE as a special case.

**Strengths:**

The paper treats cross-view links as latent variables and optimizing a likelihood bound provides a clean alternative to ad-hoc pair reweighting strategy. The design explicitly handles category-level relations (many-to-many couplings) and unalignable samples (virtual node), which prior methods largely ignore.

Experiments on Scene15, LandUse21, Caltech101, and UMPC-Food101 show consistent gains under varying mismatch and corruption rates; visualizations indicate the learned posteriors approach class-level ground truth during training.

**Weaknesses:**

1.The proposed method depends on several manually tuned hyperparameters, such as the virtual mass $\rho$ and the number of Sinkhorn iterations. The paper lacks a sensitivity analysis showing how performance varies with these values.

2.While the paper qualitatively visualizes the estimated correspondences in Figure 3, it does not provide any statistics about the intrinsic mismatch level of the datasets used. Without knowing the original category- or sample-level mismatch rate, it is difficult to judge how challenging the benchmark setting is.

3.The introduced EM-style framework, particularly the Sinkhorn-based E-step with virtual-node augmentation, likely incurs non-trivial computational overhead. The paper does not report training time or GPU memory consumption.

**Questions:**

1.Please provide a sensitivity analysis for $\rho$ and the number of Sinkhorn iterations?

2.Please report the original category-level mismatch rate in the used datasets.

3.What is the computational cost of the EM procedure? Specifically, how do the training time and GPU memory usage compare with those of DIVIDE?

---

> ### Author Response · Authors · 2025-11-25
>
> We thank the reviewer for recognizing the novelty of our approach and for the constructive feedback. In response, we have expanded our parameter analysis, provided a formal analysis of the category-level mismatch rate, and detailed the computational efficiency of our framework. We address each specific point below.
>
> > **Q1:  Sensitivity analysis for $\rho$ and Sinkhorn iteration step $t$**
>
> Thanks for the suggestion. We have included a parameter analysis for the noise rate $\rho$ and Sinkhorn iterations $t$ in **Appendix E** of the revised manuscript.
>
> We conduct experiments on the Scene15 dataset with varying $t \in [10, 1000]$ and $\rho \in [0.1, 0.5]$. The results in **Figure 6** demonstrate that CorreGen remains stable even with a small number of iterations, ensuring computational efficiency. Furthermore, under a corruption rate of 0.2, the model attains optimal performance when $\rho$ lies within the range between 0.1 and 0.2. Notably, the performance remains consistent across a broader range of values, substantiating the method's robustness to hyperparameters.
>
> > **Q2: Category-Level mismatch ratio**
>
> We thank the reviewer for this insightful suggestion. We have reported the mismatch ratio in **Appendix H** of the revised manuscript. Specifically, we i) formally derive the calculation of the Category-level Mismatch Rate (CMR), and ii) report CMR values for all datasets used in our experiments (see **Table 4**). The results indicate that these datasets universally exhibit high CMR values. Notably, we briefly analyzed the properties of the CMR metric and found that, for a fixed total dataset size, a perfectly uniform category distribution yields the highest CMR under our definition.

---

> ### Author Response · Authors · 2025-11-25
>
> > **Q3: Efficiency Concerns**
>
> Thank you for the valuable question regarding computational efficiency.  Below, we provide both theoretical complexity analysis and empirical runtime measurements of the EM procedure.
>
> Theoretically, the extra cost introduced by CorreGen mainly arises from two components in the E-step: **Sinkhorn iterations** and **GMM fitting**.
>
> - **Sinkhorn iteration**: Since the Sinkhorn iteration operates on mini-batches of fixed size $B$, its time complexity per epoch is $O(\frac{L N B}{\lambda^2})$, where $L$ is the number of Sinkhorn iterations, $N$ is the total sample size and $\lambda$ is a regularization term. The complexity grows **linearly** with the dataset size $N$. Characterized by dense matrix multiplications and element-wise operations, Sinkhorn is parallelizable and could be efficiently accelerated on GPUs.
> - **GMM**: The complexity is $O(I N  C D)$, where $I$, $C$, and $D$ denote iterations, number of clusters, and feature dimensions, respectively. GMM also scales **linearly** with the dataset size. In our experiments, to avoid numerical instability when processing high-dimensional embeddings, we implement GMM on the CPU, which introduces moderate overhead and could be accelerated by GPUs in the future.
>
> Beyond the theoretical analysis, we measured the training time and memory usage on three datasets with varying sizes. The results on the original datasets below show that although CorreGen increases training time mainly due to CPU-based GMM, GPU memory remains almost identical to DIVIDE, and overall computation stays within reasonable bounds.
>
> | Ours                            |        Scene15         |       Caltech101        |      UMPC_Food101       |
> | ------------------------------- | :--------------------: | :---------------------: | :---------------------: |
> | Size/ Batch size                |       4485/ 512        |        8677/1024        |       21695/1024        |
> | Avg epoch time                  |        1.4392s         |         5.0956s         |        13.8309s         |
> | Avg Sinkhorn iteration time     |        0.6286s         |         0.5715s         |         1.2197s         |
> | Avg GMM time                    |        0.5117s         |         3.8633s         |        10.7913s         |
> | GPU Memory (MB)                 |          524M          |          782M           |          836M           |
> | Performance (ACC \| NMI \| ARI) | 50.25 \| 48.92 \|32.87 | 68.52 \| 84.45 \| 63.45 | 49.77 \| 58.36 \| 35.73 |
>
> | DIVIDE                          |         Sceen15         |       Caltech101        |      UMPC_Food101       |
> | ------------------------------- | :---------------------: | :---------------------: | :---------------------: |
> | Size/Batch size                 |        4485/512         |        8677/1024        |       21695/1024        |
> | Avg epoch time                  |         0.2681s         |         0.6356s         |         1.4503s         |
> | GPU Memory (MB)                 |          518M           |          768M           |          834M           |
> | Performance (ACC \| NMI \| ARI) | 44.57 \| 45.98 \| 28.43 | 62.20 \| 83.30 \| 50.50 | 36.20 \| 57.92 \| 27.72 |
>
> For very large-scale datasets where GMM fitting may become a bottleneck, **CorreGen remains flexible**. As shown in our ablation study (**Table 3** in the manuscript), the E-step can be simplified by assuming a uniform marginal distribution and removing the GMM guidance.  Even under this simplified setting, the method still **significantly outperforms standard contrastive baselines**, showing that our framework scales well without compromising robustness.

---

### Author Response · Authors · 2025-12-03
**Summary of Revision**

Dear AC and SAC,

We sincerely thank you for your time and effort in evaluating our paper. Below, we provide a brief summary of the rebuttal process to assist your assessment.

We are encouraged by the positive scores **(Rating 8, 6, 6, 8)** and comments from all four reviewers. They reached a consensus regarding the value of our methodological shift from discriminative pair reweighting to principled generative modeling to address the noisy correspondence problem (**eYLV, jwwK, iE2N, and YFa9**). Specifically, **eYLV, jwwK, and YFa9** emphasized that our method effectively handles both  category-level and sample-level mismatches, **which are less explored in prior methods**. **YFa9** further highlighted our formulation as thoughtful and offering a solid rationale for robust learning. Empirically, **all reviewers** acknowledged the consistent performance gains and stability of our approach across diverse datasets with varying corruption rates.

Beyond these positive comments, we carefully addressed all concerns raised by the reviewers. Our responses are summarized below.

### Common Concerns:

- **Computation Efficiency** (**eYLV, jwwK, iE2N**):  We provided both theoretical and experimental analysis of the core computational components (i.e., Sinkhorn iterations and GMM), demonstrating that both scale linearly. Empirically, training time increases moderately due to the CPU-based GMM, while GPU memory usage remains comparable to the baseline. Furthermore, we demonstrated our framework's flexibility to adapt to large-scale scenarios while maintaining superior performance.

### Specific Concerns:

- **Comparison with OT-based methods** (**YFa9**): We evaluated the suggested baseline PROTOCOL under the same settings, and the results demonstrate that our method consistently outperforms it.
- **Generalization to multi-view scenarios** (**jwwK**): We extended experiments to the 3-view setting on Scene15. Results confirm that CorreGen effectively aggregates information.
- **Hyperparameter sensitivity analysis** (**eYLV**): We added a sensitivity analysis of the pre-defined noise rate $\rho$ and the Sinkhorn iteration step $t$ in Appendix E (Figure 6).  Results demonstrate that our method remains stable even with few iterations and maintains consistent performance across a broad range of parameter values.
- **Reliance on GMM assumptions** (**jwwK**): We clarify that our framework is flexible and compatible with general mixture models when prior knowledge is available.
- **Analysis of the posterior matrix** (**iE2N**):  We compared replacing the posterior matrix with Identity and Ground-Truth matrices, and discussed the resulting behaviors in our response. These comparisons help reveal the lower and upper performance bounds of our framework.
- **The analysis of Category-Mismatch Rate** (**eYLV**):  We have formally derived the Category-level Mismatch Rate (CMR) and added these statistics for all datasets in Appendix H (Table 4).
- **Convergence analysis** (**YFa9**): We added a formal convergence analysis in Appendix G and provided training loss curves (Figure 7).
- **Discussion of recent related works** (**YFa9**): We have updated the Related Work section to include recent advances in Incomplete Multi-view Learning and Partially View-aligned Problem as suggested.

For full details, please refer to our point-by-point responses.



Best Regards,

Authors of Paper 2596

---

### Meta-Review · Area_Chair_FS4d · 2025-12-27

**Summary:**

This paper addresses noisy correspondence in real-world multi-view clustering. The authors identify category-level mismatch and sample-level mismatch. The paper proposes a generative framework based on maximum likelihood estimation and optimized via an Expectation-Maximization algorithm. The proposed method consistently outperforms state-of-the-art baselines under varying mismatch and corruption rates. Based on the unanimous positive scores from all the reviewers, I recommend accepting this paper.

**Reviewer Concerns:**

All the concerns have been addressed well by the rebuttal.

**Reviewer Scores:**

All the reviewers gave positive initial scores and did not propose to lower the scores after the rebuttal.

---

### Decision · Program_Chairs · 2026-01-26

Accept (Oral)